# Reduced *SKP2* Expression Adversely Impacts Genome Stability and Promotes Cellular Transformation in Colonic Epithelial Cells

**DOI:** 10.3390/cells11233731

**Published:** 2022-11-22

**Authors:** Nicole M. Neudorf, Laura L. Thompson, Zelda Lichtensztejn, Tooba Razi, Kirk J. McManus

**Affiliations:** 1CancerCare Manitoba Research Institute, Winnipeg, MB R3E 0V9, Canada; 2Department of Biochemistry and Medical Genetics, Rady Faculty of Health Sciences, University of Manitoba, Winnipeg, MB R3E 0V9, Canada

**Keywords:** chromosome instability, colorectal cancer, F-box protein, SCF complex, *SKP2*, quantitative imaging microscopy

## Abstract

Despite the high morbidity and mortality rates associated with colorectal cancer (CRC), the underlying molecular mechanisms driving CRC development remain largely uncharacterized. Chromosome instability (CIN), or ongoing changes in chromosome complements, occurs in ~85% of CRCs and is a proposed driver of cancer development, as the genomic changes imparted by CIN enable the acquisition of karyotypes that are favorable for cellular transformation and the classic hallmarks of cancer. Despite these associations, the aberrant genes and proteins driving CIN remain elusive. *SKP2* encodes an F-box protein, a variable subunit of the SKP1-CUL1-F-box (SCF) complex that selectively targets proteins for polyubiquitylation and degradation. Recent data have identified the core SCF complex components (*SKP1*, *CUL1*, and *RBX1*) as CIN genes; however, the impact reduced *SKP2* expression has on CIN, cellular transformation, and oncogenesis remains unknown. Using both short- small interfering RNA (siRNA) and long-term (CRISPR/Cas9) approaches, we demonstrate that diminished *SKP2* expression induces CIN in both malignant and non-malignant colonic epithelial cell contexts. Moreover, temporal assays reveal that reduced *SKP2* expression promotes cellular transformation, as demonstrated by enhanced anchorage-independent growth. Collectively, these data identify *SKP2* as a novel CIN gene in clinically relevant models and highlight its potential pathogenic role in CRC development.

## 1. Introduction

Genome instability is an enabling characteristic of cancer characterized by an increased abundance of mutations, translocations, and/or insertions/deletions that can impact gene copy numbers and gene expression [1,2]. Collectively, these alterations may lead to the aberrant regulation, abundance, or encoded function of key genes (tumor suppressor genes, DNA repair genes, oncogenes, etc.) that may promote cancer development through the acquisition of various hallmarks associated with cancer [2]. Chromosome instability (CIN) is a prevalent form of genome instability that is defined as an increase in the rate at which whole chromosomes or chromosome fragments are gained or lost [3]. CIN is proposed to be a pathogenic event in cancer, as it drives ongoing genetic and cell-to-cell heterogeneity that may confer growth and survival advantages underlying intra-tumoral heterogeneity, cellular transformation, metastasis, drug resistance, and poor patient outcomes [1,2,4,5,6,7,8,9]. Moreover, CIN is highly prevalent in many cancer types, including colorectal cancer (CRC), where it occurs in up to 85% of all cases [1,2]. Despite these associations and their clinical implications, the molecular determinants (i.e., aberrant genes, proteins, and pathways) giving rise to CIN remain poorly understood. Accordingly, studies aimed at identifying and characterizing the aberrant genetics driving CIN will shed novel insight into CRC pathogenesis that is essential to develop novel therapeutic strategies to ultimately improve the lives and outcomes of CRC patients.

Emerging clinical and genetic data now show that aberrant expression and function of the SKP1-CUL1-F-box (SCF; S-Phase Kinase Associated Protein 1 [SKP1] and Cullin 1 [CUL1]) complex induces CIN that may have pathogenic implications in various cancer contexts [10,11,12,13,14,15]. The SCF complex is an E3 ubiquitin ligase that polyubiquitinates protein substrates to label them for proteolytic degradation via the 26S proteasome (reviewed in [12]). More specifically, the SCF complex regulates the abundance of numerous proteins, including cell cycle regulators (e.g., P27 [CDKN1B; Cyclin-dependent Kinase Inhibitor 1B] and Cyclin E1) [16,17] and transcription factors [18,19,20,21] that normally function to preserve genome stability [12,13,14,15,22,23]. The SCF complex is a quaternary protein complex comprised of three invariable core members (RBX1 [Ring-Box 1], CUL1, and SKP1) and one of 69 variable F-box proteins (e.g., SKP2; S-Phase Kinase Associated Protein 2) that impart target substrate specificity to the complex [20]. We recently identified *SKP1*, *CUL1*, and *RBX1* as novel CIN genes [13,14,15], as their reduced expression coincides with significant increases in CIN phenotypes and Cyclin E1 abundance. Cyclin E1 is an established oncogene that is genomically amplified in numerous cancer types, and its overexpression corresponds with cell cycle mis-regulation, genome instability, cellular transformation, and tumor formation in mice [16,17,24,25,26,27,28,29,30,31]. These findings support the possibility that reduced expression and/or function of key SCF complex members, including the variable F-box proteins, contributes to cancer pathogenesis. In this regard, SKP2 targets Cyclin E1 for degradation [16,32]; however, the impact reduced *SKP2* expression has on CIN and cellular transformation remains unknown.

SKP2 exhibits a role in regulating cell cycle dynamics and DNA replication by modulating the abundance of various proteins including P27 and Cyclin E1 [16,17,18,28,33]. Appropriate proteolytic targeting by SKP2 is essential for the fidelity of many key biological processes required for genome stability (e.g., cell cycle progression, signal transduction, and gene expression), and thus, it is not surprising that *SKP2* overexpression is traditionally implicated in cancer pathogenesis [17,34,35]. In this regard, *SKP2* amplification and overexpression increase P27 targeting, resulting in its reduced abundance. P27 is a cell-cycle-regulating protein that normally inhibits the G1/S phase transition and whose diminished expression is associated with disease progression, poor patient response to therapy, and worse patient outcomes in many cancer types [34,35,36,37]. Based on these observations, *SKP2* is classically described as an oncogene. Paradoxically, however, others have shown that reduced *SKP2* expression leads to P27 accumulation and is associated with mitotic defects and polyploidy [17,28,38,39,40], suggesting *SKP2* may also exhibit a tumor suppressive role. Similarly, SKP2 targets Cyclin E1 [24,25,26,41,42], whose aberrant accumulation leads to cell cycle and apoptotic defects that also promote cancer development and progression [13,14,15,24,27,43]. Thus, reduced *SKP2* expression may adversely impact SCF complex function and underlie the accumulation of protein substrates whose increased abundance promotes CIN and contributes to cancer pathogenesis [12,13,14,15,23]. Surprisingly, however, the impact reduced *SKP2* expression has on CIN, cellular transformation, and its potential impact on disease pathogenesis has yet to be determined in a CRC context.

To determine the impact reduced expression of individual F-box proteins has on CIN, we performed an siRNA-based screen of all 69 F-box proteins in which *SKP2* was identified as a strong candidate CIN gene and prioritized for subsequent study. Using a combination of bioinformatic, genomic, and single-cell quantitative imaging microscopy (QuantIM) approaches, we first determined the clinical impact of *SKP2* copy number losses and subsequently determined the impact reduced expression has in CIN, cellular transformation, and CRC pathogenesis. Using publicly available TCGA (The Cancer Genome Atlas) data [31], we show that *SKP2* copy number losses (i.e., shallow deletions) are frequent in many common cancer types and correspond with reduced mRNA expression and worse progression-free survival in CRC patients. *SKP2* silencing experiments in both malignant and non-malignant colonic epithelial cell contexts revealed that reduced *SKP2* expression corresponded with increases in CIN phenotypes, including nuclear areas, micronucleus formation and aberrant chromosome numbers. To determine the long-term impact reduced *SKP2* expression has on CIN, we generated heterozygous and homozygous *SKP2* knockout clones in which QuantIM was used to assess CIN over a 10-week period. In agreement with the siRNA-based findings, *SKP2* loss induced significant and dynamic changes in nuclear areas and chromosome counts that corresponded with cellular transformation. Thus, our findings show that *SKP2* expression is essential to preserve genome stability and thus identify *SKP2* as a novel CIN gene. They also reveal that *SKP2* loss corresponds with cellular transformation, which supports a tumor-suppressive role. Collectively, these findings, coupled with our clinical observations, are consistent with *SKP2* copy number losses and reduced expression being contributing factors in CRC pathogenesis.

## 2. Materials and Methods

### 2.1. Cell Lines and Culture

To evaluate CIN within a CRC context, we purposefully chose three karyotypically stable colonic epithelial cell lines in which to evaluate the impact of reduced *SKP2* expression on CIN phenotypes. The human (male) malignant CRC cell line HCT116 (modal chromosome number = 45) was purchased from American Type Culture Collection (Rockville, MD, USA), while two non-malignant human (male) colonic epithelial cell lines, 1CT and its derivative cell line, A1309, were generously provided by Dr. J. Shay (University of Texas Southwestern Medical Center, Dallas, TX, USA) [44,45]. HCT116 is a microsatellite instability cell line that contains a *MutL Homolog 1* (*MLH1*) deficiency underlying defects in DNA mis-match repair [46]. Thus, these cells exhibit a mutator phenotype that may produce mutations capable of synergizing with or amplifying CIN phenotypes that arise following reduced *SKP2* expression. 1CT and A1309 (modal chromosome number = 46) are non-transformed cell lines immortalized with hTERT (human telomerase reverse transcriptase) and CDK4 (Cyclin-Dependent Kinase 4); A1309 was also engineered to express mutant KRAS^G12V^ (Kirsten Rat Sarcoma Viral Proto-Oncogene), have reduced *TP53* (Tumor Suppressor Protein P53), and express a short form of *APC* (Adenomatous Polyposis Coli) truncated at residue 1309 [44,45]. To our knowledge, there are no malignant colonic epithelial cell lines that are both microsatellite stable and karyotypically stable. HCT116 cells were cultured in modified McCoy’s 5A medium (Cytiva HyClone, Vancouver, BC, Canada) supplemented with 10% fetal bovine serum (Sigma-Aldrich, Oakville, ON, Canada), while 1CT and A1309 cells were cultured in Dulbecco’s Modified Eagle Medium with high glucose/medium 199 (Cytiva HyClone, Vancouver, BC, Canada) supplemented with 2% cosmic calf serum (Cytiva HyClone, Vancouver, BC, Canada). HCT116 cells were grown in a humidified incubator at 37 °C with 5% CO_2_, whereas 1CT and A1309 cells were maintained in low-oxygen chambers containing 2% O_2_, 7% CO_2_, and 91% nitrogen in a 37 °C incubator. All cell lines were authenticated on the basis of protein expression and karyotypic analyses [11]. Additionally, all three cell lines, and HCT116 in particular, have been employed in a number of previous CIN-based studies [4,11,15,47,48,49,50,51].

### 2.2. siRNA-Based F-Box Protein Screen

The siRNA-based F-box protein screen was performed in duplicate (number of biological replicates [N] = 2) using a similar approach to that detailed previously [15]. Briefly, custom-arrayed, 96-well reverse transfection format plates were purchased from Dharmacon (Horizon Discovery Biosciences Ltd., Cambridge, UK). HCT116 cells were seeded into each well containing rehydrated siRNAs and DharmaFECT mixture. Cells were permitted to grow for 4 days at 37 °C at which point they were fixed (4% paraformaldehyde; 10 minutes [min]), counterstained (Hoechst 33342; Sigma-Aldrich, Oakville, ON, Canada), and subjected to QuantIM for changes in nuclear areas as detailed elsewhere [4,52].

### 2.3. SKP2 Copy Number Alterations, Reduced Expression, and Survival Analyses

Publicly available genomic (gene copy number) and clinical data were extracted from TCGA Pan-Cancer Atlas [31] for eight common cancer types (bladder; breast; CRC; glioblastoma; lung; ovarian; pancreatic; prostate) as detailed elsewhere [11]. Survival data were imported into Prism 9 (GraphPad, San Diego, CA, USA), stratified by gene either copy number losses, copy number gains, or mutations, and Kaplan–Meier (KM) survival curves were generated and statistically compared using log-rank tests with a *p*-value < 0.05 considered statistically significant.

### 2.4. Silencing and Western Blot Analyses

*SKP2* silencing was performed using RNAiMAX (Life Technologies, Burlington, ON, Canada) and ON-TARGETplus siRNA duplexes (Dharmacon, Horizon Discovery Biosciences Ltd., Cambridge, UK). Four individual siRNA duplexes targeting distinct coding regions of *SKP2* mRNA (siSKP2-1, -2, -3 or -4) or a pooled siRNA (siSKP2-Pool) comprised of equal molar amounts of each individual siRNA were used, along with a non-targeting siRNA (siControl). Silencing efficiencies were determined using semi-quantitative western blot analyses four days post-transfection [53], using antibodies and dilutions specified in Appendix A. Semi-quantitative image analyses were employed to determine relative protein expression levels in which SKP2 abundance (band intensities) were normalized to the corresponding loading control (Cyclophilin B) and are presented relative to siControl in the siRNA-based experiments or a non-targeting-control (NT-Control) for the CRISPR/Cas9 clone experiments.

### 2.5. QuantIM and CIN Analyses

QuantIM approaches were employed to assess changes in CIN-associated phenotypes, including changes in nuclear areas and micronucleus formation, as detailed previously [4,54]. Briefly, cells were seeded in 96-well optical bottom plates 24 h in advance, silenced in sextuplet (number of technical replicates [n] = 6), and permitted to grow for 4 days, whereupon they were fixed (4% paraformaldehyde), stained (Hoechst 33342), and imaged using a Cytation 3 Cell Imaging MultiMode Reader (BioTek, Winooski, VT, USA). Nuclear areas and micronucleus formation were automatically quantified using Gen5 (BioTek, Winooski, VT, USA) software, as detailed elsewhere [4,14,52,54]. All quantitative data were imported into Prism, where descriptive statistics and non-parametric tests were performed, including two-sample Kolmogorov–Smirnov (KS) tests comparing cumulative nuclear area distribution frequencies and Mann–Whitney (MW) tests assessing differences in the rank order of micronucleus formation frequencies, where *p*-values < 0.05 are considered significant. All graphs were generated and assembled in Prism. Experiments were performed in triplicate (N = 3).

### 2.6. Mitotic Chromosome Spread Enumeration

Mitotic chromosome spreads were generated as detailed elsewhere, with brief (<2 h) Colcemid treatments used to enrich mitotic populations [52,53]. A minimum of 100 spreads per condition were enumerated, with all experiments performed in triplicate, except for the temporal *SKP2* clone studies in which each clone was assessed once at every timepoint. Student’s *t*-tests were employed to identify statistically significant differences in the total frequencies of aberrant chromosome numbers in *SKP2* silenced cells relative to siControl for all biological replicates.

### 2.7. CRISPR/Cas9 Approaches

*SKP2* knockout clones were generated using a two-step CRISPR/Cas9 approach in A1309 cells with *SKP2*-targeting and non-targeting control synthetic guide RNAs (sgRNAs) according to the manufacturer (Sigma Aldrich, St. Louis, MO, USA) and as detailed previously [13]. Briefly, A1309 cells were transduced with lentivirus particles containing two distinct *SKP2* synthetic guide RNAs (sgRNAs) or a NT-Control (Appendix A) that co-expresses blue fluorescent protein (BFP). Cells were sorted by fluorescence-activated cell sorting (FACS), with transduced populations (BFP+) isolated and subsequently transfected with a plasmid that co-expresses Cas9 and green fluorescent protein (GFP). FACS was used to recover BFP+/GFP+ cells, with individual clones isolated through serial dilutions. Putative *SKP2* knockout clones with reduced expression were identified by Western blot, while DNA sequencing was employed to identify the allele-specific edits (Génome Quebec, Montreal, QC, Canada).

### 2.8. Soft Agar Colony Formation Assay

Three-dimensional colony formation assays were performed as described previously [52,55]. Briefly, 20,000 cells/well were combined with 0.4% agar and seeded into a six-well plate containing a base layer of 0.6% agar, with HCT116 cells serving as a positive control [52]. Cells were supplemented with media replaced every week for 4 weeks, at which point cells were fixed (4% paraformaldehyde), stained (0.005% crystal violet), and imaged using a Cytation 3 equipped with a 4× objective. Gen5 software was employed to enumerate colonies, with colonies being operationally defined as those with a diameter ≥100 μm in size. Experiments were performed twice.

## 3. Results

### 3.1. An siRNA-Based F-Box Protein Screen Identifies SKP2 as a Strong Candidate CIN Gene

We previously identified the three core SCF complex members as novel CIN genes [13,14,15]; however, the specific contributions individual F-box proteins have in CIN, and CRC pathogenesis remains largely unexplored. To gain initial insight into the impact reduced F-box protein expression may have on CIN, we performed a comprehensive siRNA-based screen of all 69 F-box proteins to identify those with the greatest impacts on CIN. Briefly, each F-box protein was individually silenced in karyotypically stable HCT116 cells and QuantIM was employed to identify significant changes in nuclear areas relative to a non-silencing control (siControl). Conceptually, changes in nuclear areas are used as a surrogate marker of CIN, with increases or decreases typically associated with gains or losses in chromosome numbers, respectively [5,56,57]. Of the 69 genes screened, the silencing of 64 genes corresponded with significant differences in cumulative nuclear area distribution frequencies relative to siControl, with 8 exhibiting significant decreases and 56 exhibiting significant increases (Figure 1). Notably, *FBXO5* silencing induced the greatest increases in nuclear areas; however, silencing was associated with visual decreases in cell numbers (i.e., death) and agrees with *FBXO5* being an essential gene as indicated within the Cancer Dependency Map (DepMap; https://depmap.org/; accessed on 29 September 2022) [58], whereas *SKP2* silencing corresponded with the second largest increases but had minimal impact on cell numbers. Moreover, SKP2 has established proteolytic targets such as P27 and Cyclin E1 [16,17,28,41], which are oncogenes frequently amplified at the level of the genome in many cancers [29,30,31], including ~20% and ~23% of CRC cases, respectively [31]. Accordingly, we predicted that reduced *SKP2* expression would prevent P27 or Cyclin E1 degradation leading to its increased abundance that would effectively phenocopy genomic amplification and induce CIN and cellular transformation. Thus, *SKP2* was purposefully selected for subsequent in-depth analyses.

### 3.2. SKP2 Copy Number Losses Are Present in CRC and Correspond with Reduced Expression and Worse Patient Outcomes

To determine the potential clinical impact reduced *SKP2* expression may have in cancer, TCGA data [31] from eight common cancer types (see Materials and Methods) were assessed for *SKP2* copy number losses. Figure 2A shows that while deep (homozygous) deletions are relatively rare (0–0.2% of all cancer cases), shallow (heterozygous) deletions are more prevalent and range from 3.1% (15/489 cases) to 10.8% (63/592 cases) in pancreatic cancer and CRC, respectively. Given that ~175,000 North Americans are diagnosed with CRC each year [59,60] it is estimated that ~18,400 will have tumors harboring *SKP2* copy number losses. Moreover, and in agreement with a potential pathogenic role, *SKP2* copy number losses correspond with significant decreases in mRNA expression in CRC patients (Figure 2B). Finally, to determine the potential clinical impact *SKP2* copy number losses may have for CRC patients, a KM curve was generated that revealed statistically worse progression-free survival for patients with copy number losses relative to those with diploid copy numbers (Figure 2C) [29,30,31]. Although not a focus of the current study, similar health outcome analyses were performed for CRC patient samples with copy number gains and mutations. No statistical differences (*p*-values > 0.05) in survival outcomes were observed when comparing cases with either *SKP2* copy number gains or *SKP2* mutations relative to diploid cases [31]. Thus, based on these analyses, only *SKP2* copy number losses are associated with statistically worse outcomes. Collectively, these findings are consistent with *SKP2* copy number losses and reduced expression contributing to CRC pathogenesis and therefore warrant further molecular investigations.

### 3.3. Diminished SKP2 Expression Induces Increases in CIN-Associated Phenotypes in HCT116 Cells

The above data coupled with our previous studies identifying the invariable core SCF complex members as novel CIN genes [13,14,15] support the possibility that reduced *SKP2* expression may also induce CIN. Prior to assessing the impact reduced *SKP2* expression has on CIN, we first assessed the silencing efficiencies of individual (siSKP2-1, -2, -3, -4) and pooled (siSKP2-Pool) siRNA duplexes in HCT116, a karyotypically stable CRC cell line. As shown in Figure 3A (Appendix A), semi-quantitative western blots identified siSKP2-3 and -4 as the most efficient individual siRNAs, which along with the siSKP2-Pool were employed in all subsequent experiments as they typically reduced SKP2 protein levels to <10% of siControl. Moreover, western blotting also revealed that *SKP2* silencing corresponded with increases in P27 abundance, indicating that reduced *SKP2* expression adversely impacted normal SCF^SKP2^ function and proteolytic targeting of P27 (Appendix A).

To determine the impact reduced *SKP2* expression has on CIN, QuantIM was employed to statistically assess changes in nuclear areas and micronucleus formation relative to siControl. QuantIM is an established approach capable of rapidly assessing the cell-to-cell heterogeneity in aberrant CIN phenotypes, including changes in nuclear areas and micronucleus formation [5,54]. While changes in nuclear areas are typically associated with large-scale changes in chromosome complements (i.e., polyploidy) [13,61], micronuclei are extranuclear bodies that frequently arise from chromosome mis-segregation events and are hallmarks of CIN [62,63]. As predicted, reduced *SKP2* expression induced visual increases in nuclear area sizes and heterogeneity relative to siControl (Figure 3B) that correspond with significant increases in cumulative nuclear area distribution frequencies (Figure 3C, Appendix A). By contrast, *SKP2* silencing did not induce significant increases in micronucleus formation in HCT116 (Figure 3D,E, Appendix A). Collectively, these data corroborate those of the initial F-box protein screen and suggest *SKP2* is a novel CIN gene in a malignant CRC context.

**Figure 3 cells-11-03731-f003:**
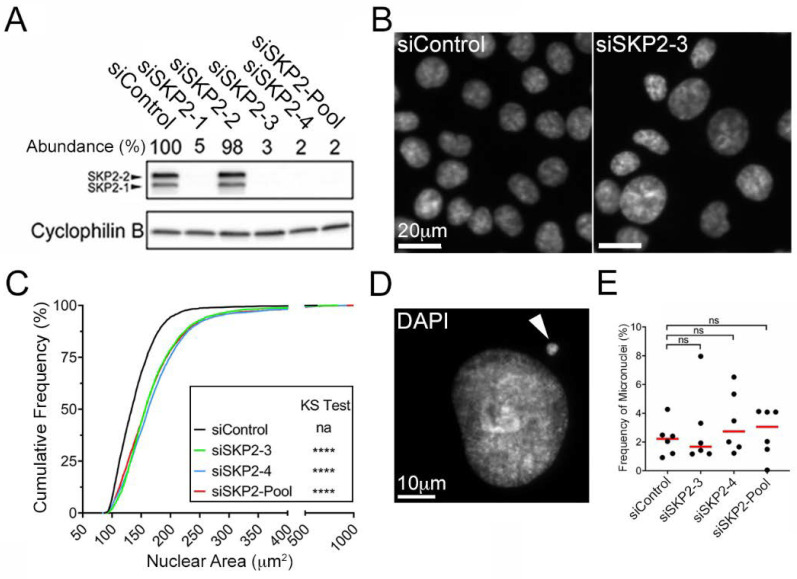
*SKP2* Silencing Corresponds with Significant Increases in Nuclear Areas in HCT116. (**A**) Western blots presenting SKP2 abundance following siRNA-based silencing with individual (siSKP2-1, -2, -3, -4; top labels) and pooled (siSKP2-Pool) siRNAs relative to siControl; Cyclophilin B serves as the loading control. Note that two bands representing the two SKP2 isoforms (SKP2-1 and SKP2-2; left labels) are visible [64,65]. Semi-quantitative analyses were performed whereby SKP2 abundance was first normalized to the respective loading control and is presented relative to siControl (100%). (**B**) Low-resolution images of Hoechst-counterstained nuclei showing visual increases in nuclear areas following *SKP2* silencing relative to siControl. (**C**) Cumulative distribution frequency graph reveals significant increases (rightward shift) in nuclear areas following *SKP2* silencing relative to siControl (two-sample KS test; na, not applicable; ****, *p*-value < 0.0001; N = 3, n = 6). (**D**) High-resolution image presenting a Hoechst-counterstained nucleus and associated micronucleus (arrowhead). (**E**) Dot plot and subsequent statistical analyses (MW test; ns, not significant, *p*-value > 0.05) fail to identify significant changes in micronucleus formation following *SKP2* silencing (red bars identify median values; N = 3, n = 6).

### 3.4. SKP2 Is Required to Maintain Chromosome Stability in HCT116 Cells

While the QuantIM approaches employed above assess CIN phenotypes, they do not specifically quantify changes in chromosome numbers. To determine the impact reduced *SKP2* expression has on chromosome numbers, mitotic chromosome spreads were generated, enumerated, and statistically compared with siControl. As HCT116 cells have a modal number of 45 chromosomes, aberrant spreads were classified into one of three categories (Figure 4A): (1) losses (<45 chromosomes); (2) small-scale gains (46–54 chromosomes); or (3) large-scale gains (≥55 chromosomes). In general, deviations from the modal chromosome number occurred following *SKP2* silencing that included both losses and gains. More specifically, *SKP2* silencing induced a 1.6- to 2.0-fold increase in the total number of aberrant spreads relative to siControl (Figure 4B), with the greatest changes typically being chromosome losses and small-scale gains. Moreover, the total abundance of aberrant spreads across all biological replicates ranged from 1.8- to 2.1-fold increases (Figure 4C, Appendix A). Interestingly, further scrutiny of the images (Figure 4A; bottom right) revealed that ~90% of the chromosome spreads harboring large-scale gains exhibited cytological features consistent with endoreduplication (i.e., pairs of paired sister chromatids) or an extra round of DNA replication in the absence of cytokinesis [66]. Collectively, these findings show that reduced *SKP2* expression induces CIN in HCT116 cells.

### 3.5. Reduced SKP2 Expression Induces CIN in Non-Malignant, Non-Transformed Colonic Epithelial Cell Contexts

Although the above findings identify *SKP2* as a novel CIN gene in a malignant CRC context, the impact reduced *SKP2* expression has in models of early disease development remains unknown. Accordingly, QuantIM analyses were performed in two clinically relevant models representing early disease states. 1CT and its derivative line, A1309, are karyotypically stable, non-malignant/non-transformed colonic epithelial cell lines [45] and each has a modal chromosome numbers of 46 [11]. Moreover, while 1CT and A1309 are immortalized with hTERT and CDK4, A1309 also harbors reduced TP53 expression and expresses mutant forms of KRAS (KRAS^G12V^) and APC truncated at amino acid residue 1309 [45].

Prior to performing the CIN assays, western blots were performed and determined that the silencing efficiencies of the individual (siSKP2-3 and -4) and pool (siSKP2-Pool) siRNAs were highly effective within both lines and generally reduced expression to <10% of siControl levels (Figure 5A and Appendix A), which corresponded with P27 accumulation in A1309 cells (Appendix A). In agreement with the HCT116 findings, the cumulative nuclear area distribution frequencies revealed statistically significant increases relative to siControl in both lines but were more pronounced within the A1309 cells (Figure 5B, Appendix A). Moreover, *SKP2* silencing failed to induce significant increases in micronucleus formation in 1CT cells but did induce significant increases in A1309, specifically for the SKP2-4 and SKP2-Pool conditions (Figure 5C, Appendix A). Next, chromosome enumeration in 1CT cells revealed 2.9–4.0-fold increases in the frequency of aberrant chromosome spreads that were determined to be significant (Figure 5D,E, Appendix A) within the siSKP2-4 condition, with the siSKP2-3 and siSKP2-Pool conditions trending towards significance (*p*-values = ~0.10). Additionally, there was a similar, albeit less pronounced 1.8–2.7-fold increase in aberrant chromosome spreads in A1309 (Figure 5D,E, Appendix A), with siSKP2-3 and siSKP2-4 attaining statistical significance and siSKP2-Pool trending towards significance (*p*-value = 0.06). It should also be noted that there is an ~3-fold difference in the baseline frequency of aberrant spreads observed within the siControl conditions between the cell lines, with 1CT and A1309 harboring ~8% and ~24% aberrant spreads, respectively. Like the HCT116 data, a large proportion (~50%) of the aberrant spreads from both cell lines categorized with large-scale gains exhibited hallmarks of endoreduplication. Collectively, these findings show that reduced *SKP2* expression induces CIN phenotypes in non-transformed cells that are generally more prevalent within A1309 cells. Thus, these findings identify *SKP2* as a novel CIN gene in two non-malignant/non-transformed colonic epithelial cell models of early disease development.

### 3.6. SKP2 Loss Corresponds with Dynamic CIN Phenotypes in Models of Disease Development

Having determined that *SKP2* copy number losses occur in 10.8% of CRC patients and that reduced expression corresponds with CIN, which is proposed to be an early driver of CRC development, we next sought to develop clinically relevant heterozygous and homozygous models in which CIN could be assessed over time. This is particularly important as CIN is a dynamic phenotype that contributes to ongoing cell-to-cell and genetic heterogeneity. Accordingly, CRISPR/Cas9 was used to produce one heterozygous (*SKP2*^+/−^1) and three homozygous clones (*SKP2*^−/−^A; *SKP2*^−/−^B; *SKP2*^−/−^C) in A1309 cells. A1309 cells were purposefully selected as they contain genetic alterations associated with early disease development that appear to synergize with *SKP2* loss, as evidenced by the more pronounced CIN phenotypes in the preceding section. Western blots and DNA sequencing were used to confirm reduced expression and identify the allele-specific edits (Appendix A). In particular, *SKP2*^+/−^1 exhibited expression levels that are ~30% of NT-Control, while none of the *SKP2*^−/−^ clones expressed any detectable SKP2 (Appendix A). Western blotting also revealed that the *SKP2* clones exhibited increases in P27 abundance relative to NT-Control (Appendix A). Next, DNA sequencing revealed that SKP2^+/−^1 harbors a single base pair (bp) deletion in one allele, whereas *SKP2*^−/−^A has a 2 bp deletion in allele 1 and a 4 bp deletion in allele two, *SKP2*^−/−^B incorporated a 14 bp and a 1 bp deletion, while *SKP2*^−/−^C harbors a 1bp and a 2 bp deletion (Appendix A). In all instances, the allele-specific edit(s) are predicted to introduce premature stop codons (Appendix A) that are expected to induce nonsense-mediated mRNA decay, which is supported by the lack of small molecular weight protein products (Appendix A).

To determine the temporal impact heterozygous and homozygous loss of *SKP2* has on CIN, each *SKP2* and NT-Control clone was continually passaged for 10 weeks with cellular aliquots subjected to QuantIM analyses every four passages (p), or approximately every 2 weeks. In agreement with the siRNA data, CIN was both prevalent and dynamic in each *SKP2* clone (Figure 6). More specifically, heterozygous and homozygous loss of *SKP2* corresponded with ongoing and dynamic changes in cumulative nuclear area distributions. Interestingly, *SKP2*^+/−^1 exhibited the most dynamic changes in cumulative nuclear area distributions of all the *SKP2* clones investigated. For example, *SKP2*^+/−^1 exhibited significant increases in overall distributions at p0 and p12, significantly smaller distributions at p4 and p8, and similar distributions at p16 and p20 (Figure 6A, Appendix A). On the other hand, *SKP2*^−/−^A exhibited significant increases at all time points except for p4, when the distribution was more like the NT-Control but was still statistically distinct. *SKP2*^−/−^B displayed similar trends to that of *SKP2*^−/−^A but exhibited more dramatic increases with each successive passage, except for p16. While each *SKP2* clone displayed dynamic increases and/or decreases in nuclear area distributions over the time course, *SKP2*^−/−^A and *SKP2*^−/−^B consistently exhibited the largest increases. In contrast, *SKP2*^−/−^C exhibited more subtle differences, with significant increases observed at each passage except for p20, when it was significantly decreased.

Next, micronucleus formation was statistically compared between *SKP2* and NT-Control clones. As shown in Figure 6B, there were dynamic and significant changes in micronucleus formation over time (Appendix A). For example, *SKP2*^+/−^1 presented significant increases at p0, p4, p8, and p16 but showed trending but not significant increases at p12 and p20. With respect to the homozygous clones, *SKP2*^−/−^A exhibited significant increases at p4, p8, p12, and p20 and trending increases at p0 and p16, while *SKP2*^−/−^B exhibited the largest increases of any clone at each timepoint, whereas *SKP2*^−/−^C only exhibited significant increases at p4, p8, and p16. Overall, the *SKP2* clones all exhibit dynamic and significant changes in micronucleus formation over the 10-week time course.

Finally, to determine the impact *SKP2* loss has on chromosome complements, mitotic chromosome spreads were generated and assessed at each timepoint (Figure 6C, Appendix A). In general, each *SKP2* clone exhibited increases in the total number of aberrant spreads relative to NT-Control that were typically greatest at p0 (3.8–4.9-fold relative to NT-Control) and remained dynamic but tended to decrease over time (p20; 0.9–1.8-fold). For example, *SKP2*^+/−^1 exhibited a 4.9-fold increase in aberrant spreads that decreased to 0.5-fold at p12 before increasing to 3.1- and 1.5-fold at p16 and p20, respectively, while the homozygous clones also exhibited similar dynamics over time (Figure 6C). Collectively, the dynamic and significant changes in nuclear areas and micronucleus formation, coupled with the ongoing changes in chromosome complements, confirm *SKP2* as a novel CIN gene in non-malignant/non-transformed colonic epithelial cells and further suggest that *SKP2* copy number losses may be an early etiological event contributing to CRC development.

### 3.7. SKP2 Loss Promotes Anchorage-Independent Growth and Cellular Transformation

As CIN is proposed to be an early etiological event contributing to cellular transformation and disease pathogenesis [67,68], we next sought to determine whether *SKP2* loss promotes anchorage-independent growth, a key indicator of cellular transformation. Accordingly, 3D soft agar assays (Figure 7A,B) were employed for each *SKP2* and NT-Control clone, with HCT116 cells serving as a positive control [69]. While there was no evidence of an increase in colony formation for the heterozygous (*SKP2*^+/−^) clone, homozygous loss was associated with increases in colony numbers. More specifically, there was a 2.2–4.0-fold increase in colony formation for the *SKP2*^−/−^ clones (Figure 7C). Accordingly, homozygous loss of *SKP2* promotes anchorage-independent growth and thus underlies cellular transformation, further supporting the possibility that *SKP2* loss may contribute to early disease development.

## 4. Discussion

In this study, we employed complementary QuantIM approaches to determine the impact reduced *SKP2* expression has on CIN and its potential implications for early CRC development in both short- and long-term assays. To begin, in silico analyses of TCGA data revealed that *SKP2* copy number losses occur in 10.8% of CRC cases, are associated with reduced expression (mRNA), and correspond with worse progression-free survival, while worse survival outcomes were not observed for patient samples exhibiting *SKP2* copy number gains or mutations. To functionally evaluate the impact reduced *SKP2* expression has on CIN, transient silencing was performed that induced significant increases in nuclear areas and chromosome complements in both malignant (HCT116) and non-malignant/non-transformed (1CT and A1309) cells, with significant increases in micronucleus formation also observed in A1309 cells. Next, we generated heterozygous and homozygous *SKP2* clones in non-malignant, non-transformed A1309 cells and assessed CIN at regular intervals over a 10-week period. In agreement with a pathogenic role in early disease development, heterozygous and homozygous loss of *SKP2* were induced dynamic and significant changes in nuclear areas, micronucleus formation, and aberrant mitotic chromosome spreads, while homozygous loss also promoted cellular transformation as demonstrated by enhanced anchorage-independent growth (i.e., colony formation). Collectively, these findings identify *SKP2* as a novel CIN gene in both malignant and non-malignant colonic epithelial contexts and further suggest that *SKP2* loss and/or reduced expression may be a significant yet underappreciated driver of early disease development.

As CIN reflects both gains and losses of whole chromosomes and/or chromosome fragments [1], it is expected that reduced *SKP2* expression would induce both increases and decreases in chromosome complements. Interestingly, however, the QuantIM analyses discovered that diminished *SKP2* was primarily associated with significant increases in nuclear areas, whereas the chromosome enumeration studies revealed that chromosome losses were the most frequent aberrant phenotype. While the precise mechanism(s) accounting for these perceived discrepancies remains unknown, there are at least four technical and/or biological considerations that may account for this dichotomous observation. First, the nuclear area analyses are conducted exclusively on interphase populations, whereas the chromosome enumeration assays are limited to mitotic populations (e.g., prometaphase-metaphase). Conceptually, accurately enumerating chromosomes mandates that each cell within a given experimental population/condition is equally capable of entering and remaining arrested in mitosis, as cells are treated with Colcemid. Since the analyses are conducted on asynchronous populations, only a small proportion of cells, typically < 5%, are in mitosis, whereas most cells (~95%) are in interphase (G1, S-phase and G2). Thus, mitotic chromosome analysis enriches for populations of cells with a higher capacity for entering and remaining in mitosis. Second, reduced *SKP2* expression and/or diminished SCF^SKP2^ function is expected to adversely impact cell cycle progression and the number of cells entering mitosis. SCF^SKP2^ normally regulates Cyclin E1 degradation [17,70], which is a critical factor for the G1 to S-phase transition [71,72]. Consequently, reduced *SKP2* expression or function would induce aberrantly high levels of Cyclin E1 that are expected to adversely impact replication and cell cycle progression. Third, recall that the large nuclear area increases observed in cells with reduced *SKP2* expression are associated with large-scale changes in DNA content (i.e., polyploidy). A previous study by Hall et al. [73] determined that polyploid cells progress through mitosis more rapidly than diploid cells. More specifically, they showed that polyploid cells prematurely exit mitosis, which is predicted to result in fewer polyploid cells being captured for mitotic chromosome spread analyses. Finally, endoreduplication (also known as endoreplication) may also impact the frequency of mitotic chromosome spreads with increased chromosome complements. Endoreduplication is an aberrant biological process in which cells re-replicate their DNA without entering mitosis (reviewed in [70]), and its presence correlates with disease pathogenesis in many cancer types [74,75], including CRC [76,77,78]. Thus, cells undergoing endoreduplication will be underrepresented within the mitotic chromosome spreads but are included within the nuclear area analyses. Coincidentally, genomic amplification or ectopic overexpression of the Cyclin E1 gene has been shown to induce endoreduplication [14,15,27], while Nakayama et al. [16,17,28] also observed increases in nuclear areas and endoreduplication in multiple cell types isolated from *SKP2* knockout mice; however, they never specifically assessed colonic epithelial cells. As SCF^SKP2^ normally regulates Cyclin E1 degradation [16,17,41], reduced *SKP2* expression is expected to underlie increases in Cyclin E1 abundance that is predicted to phenocopy genomic amplification and induce endoreduplication. This possibility is supported by our observations that a large proportion (from ~50% to 90%) of mitotic chromosome spreads exhibiting large-scale increases in chromosome numbers display hallmarks of endoreduplication (i.e., pairs of paired sister chromatids). Observations by Thompson and colleagues further strengthen this possibility, as they noted that *SKP1* silencing corresponds with increases in Cyclin E1 abundance and endoreduplication in HCT116 cells [15]. Accordingly, the heterogeneous results we observe are likely explained by a combination of technical and/or biological considerations stemming from the various QuantIM assays employed and the aberrant biology resulting from reduced *SKP2* expression. Our findings also underscore the importance of employing multiple complementary approaches to identify and accurately assess novel CIN genes.

Although *SKP2* was identified as a novel CIN gene in CRC precursor cells, the frequencies and magnitudes of the CIN phenotypes were heterogeneous between cell lines and replicates. For example, the QuantIM analyses revealed significant increases in the frequency of aberrant chromosome spreads in all *SKP2* silenced conditions and in all three experimental replicates from the malignant HCT116 cells, whereas the non-malignant/non-transformed 1CT cells exhibited less pronounced changes that were significant with siSKP2-4. Although the molecular basis for this difference is unknown, it is plausible that the altered frequencies of aberrant spreads may be due in part to the distinct genetic contexts of the various cell lines employed. For example, HCT116 is a microsatellite instability cell line that contains a *MutL Homolog 1* (*MLH1*) deficiency underlying defects in DNA mis-match repair [46]. Thus, these cells harbor a mutator phenotype that may produce mutations capable of synergizing with reduced *SKP2* expression to enhance the various CIN phenotypes assessed. Furthermore, HCT116 also express the KRAS^G13D^ oncoprotein enabling them to proliferate independent of growth factor signaling [79]. As *SKP2* normally regulates key cell cycle regulation proteins, such as Cyclin E1 and P27, the aberrant cell cycle dynamics coupled with the enhanced proliferative signaling inherent to HCT116 cells may further exacerbate the CIN phenotypes [16,17,28]. Similarly, the supplementary genetic differences distinguishing A1309 from 1CT may also explain the enhanced phenotypes observed in A1309 cells. Recall that A1309 cells express mutant KRAS^G12V^, truncated APC, and have reduced *TP53* expression that collectively may exacerbate micronucleus formation through the synergistic deregulation of the biological pathways they normally regulate, including cell cycle progression [80,81], proliferative signaling [82], microtubule stabilization [83,84], DNA double-strand break repair [81,85] and apoptotic signaling [81,85]. It should be noted that changes in nuclear areas and increased micronucleus formation typically arise through aberrations in distinct cellular pathways, including cell cycle dysregulation or centrosome overduplication, and mitotic spindle dynamics or errors in double-strand break repair, respectively [1,2]. Importantly, it should be re-iterated that the *KRAS*, *APC,* and *TP53* alterations by themselves, do not induce micronucleus formation, as the frequency of micronuclei is similar and low (~1%) in both 1CT and A1309. Accordingly, these defects, in combination with reduced *SKP2* expression, may enable DNA damage and chromosome segregation errors to persist [45], which may account for the increased frequency of micronuclei observed in A1309 relative to 1CT cells. In any case, it is likely that the distinct genetic backgrounds coupled with the CIN phenotype, which induces heterogeneous phenotypes, collectively contribute to the variation observed within and between the various colonic cellular contexts.

*SKP2* is classically considered an oncogene, as it is amplified and overexpressed in various cancer contexts, including breast, T-cell lymphoma, melanoma, and Kaposi’s sarcoma. Moreover, its overexpression in these cancers correlates with disease progression, worse patient outcomes, and poor therapeutic response [34,35,36,37,41,86,87,88]. This oncogenic assignment stems primarily from the resulting enhanced degradation of the haploinsufficient tumor suppressor protein, P27 (reviewed in [89]), which normally functions to regulate cyclin-dependent kinase activity and oppose cell-cycle progression at the G1 to S-phase boundary [37]. In fact, Shapira et al. [40] examined SKP2 abundance in 80 CRC patient samples and determined that enhanced *SKP2* expression correlated with reduced P27 abundance, loss of tumor differentiation, and decreased overall survival. Interestingly, however, more recent studies have shown that aberrant P27 accumulation is associated with CIN and mitotic defects [38,39]. These seemingly opposing observations strongly imply that P27 abundance is tightly regulated in a cell cycle-dependent manner and that too much or too little expression can have pathogenic implications. Beyond P27, SKP2 also regulates Cyclin E1, an established oncoprotein whose overexpression leads to cell cycle defects that promote cancer development and progression [24,27,43]. Indeed, the Cyclin E1 gene is genomically amplified in many cancer types, including CRC, and its ensuing overexpression promotes cell cycle mis-regulation, CIN, cellular transformation, and tumor formation in mice [27,29,30,31]. In agreement with these observations, more recent studies have established that reduced expression of the core SCF complex members, namely *SKP1*, *CUL1*, and *RBX1,* underlies increases in Cyclin E1 protein abundance, CIN, and cellular transformation in CRC and ovarian cancer contexts [13,14,15]. Thus, the results of these previous studies and those of the current study are consistent with *SKP2* expression and function being tightly regulated to ensure accurate cell cycle progression, maintain genome stability, and prevent cancer development (i.e., tumor suppressor-like activity). Thus, it appears that *SKP2* harbors both oncogene-like and tumor-suppressor-like properties depending on the cellular context and the aberrant regulation of the substrate targets of the SCF^SKP2^ complex.

In summary, our data demonstrate that reduced *SKP2* expression induces dynamic and heterogeneous increases in nuclear areas, micronucleus formation, and aberrant chromosome numbers in various colonic epithelial cell contexts, which collectively identify *SKP2* as a novel CIN gene. They also advance our fundamental understanding of how reduced *SKP2* expression impacts CIN and cellular transformation and reveal a potential role for reduced *SKP2* expression in early disease development. As SKP2 is one of two established F-box proteins that selectively target Cyclin E1, it now becomes imperative to gain mechanistic insight into the underlying mechanism(s) contributing to CIN within each of those contexts. As the spectrum of substrate specificities for each F-box protein remains poorly characterized, proteomic and functional studies are required to determine the impact aberrant substrate accumulation has on CIN, cellular transformation, and disease development. Finally, as *SKP2* copy number losses and aberrant expression are prevalent in a myriad of cancer types, our findings likely have broad pathogenic implications beyond the CRC contexts of this study.

## Figures and Tables

**Figure 1 cells-11-03731-f001:**
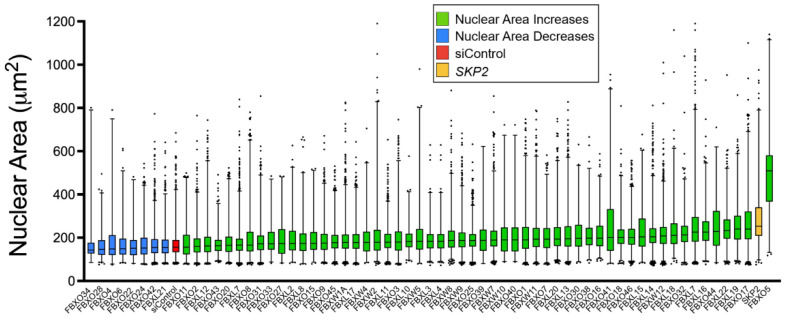
SiRNA-based Screen Identifies *SKP2* as a Strong Candidate CIN Gene. Box-and-whisker plot presenting the F-box genes that, when silenced, induce statistically significant differences (two-sample Kolmogorov–Smirnov [KS] tests) in cumulative nuclear area distribution frequencies relative to siControl. Genes are arranged by increasing median nuclear area with boxes identifying the interquartile ranges (25th, 50th, 75th percentiles) and the whiskers denoting the 1st and 99th percentiles. Blue boxes denote significant decreases in nuclear areas, while green boxes denote significant increases in nuclear areas. This screen identified *SKP2* as a strong candidate CIN gene (yellow).

**Figure 2 cells-11-03731-f002:**
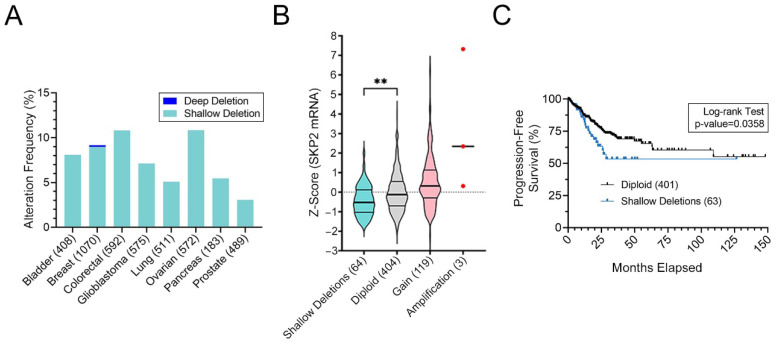
*SKP2* Copy Number Losses are Frequent in Cancer and are Associated with Reduced Expression and Worse Progression-Free Survival in CRC. (**A**) Bar graph presenting the frequency of *SKP2* copy number losses (deep [homozygous] and shallow [heterozygous] deletions) in eight common cancer types (total cases) [29,30,31]. Note that shallow deletions occur in 10.8% of CRC cases. (**B**) Violin plots reveal that *SKP2* shallow deletions in CRC correspond with significant decreases in expression (mRNA) relative to diploid controls (MW test; **, *p*-value < 0.01) [29,30,31]. Horizontal lines identify the 25th, 50th, and 75th percentiles, while the total number of cases in each category is indicated within brackets. (**C**) CRC patients with *SKP2* shallow deletions (blue) have significantly worse progression-free survival relative to patients with diploid copy numbers (black) (Log-rank test; *p*-value ≤ 0.05) [29,30,31].

**Figure 4 cells-11-03731-f004:**
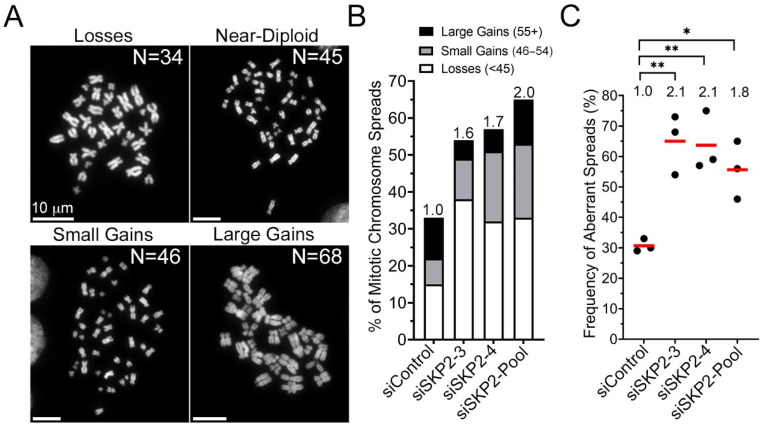
Reduced *SKP2* Expression Induces Significant Changes in Aberrant Chromosome Numbers in HCT116. (**A**) Representative high-resolution images of DAPI counterstained mitotic chromosome spreads displaying the modal number of 45 chromosomes, chromosome losses (≤44), small-scale chromosome gains (46–54), and large-scale chromosome gains (≥55), including endoreduplication (bottom right), with the total chromosome number (N) indicated in the top right corner of each image. (**B**) Bar graph reveals increases in the percentage of aberrant mitotic chromosome spreads following *SKP2* silencing relative to siControl. The fold increase in aberrant spreads is indicated at the top of each column and is presented relative to siControl (N = 3, n = 100 spreads/condition). (**C**) Dot plot presenting the frequency of total aberrant spreads following *SKP2* silencing with the relative fold increase indicated (top of each category); red bars identify mean values (Student’s *t*-test; *, *p*-value ≤ 0.05; **, *p*-value < 0.01; N = 3, n = 100 spreads/condition).

**Figure 5 cells-11-03731-f005:**
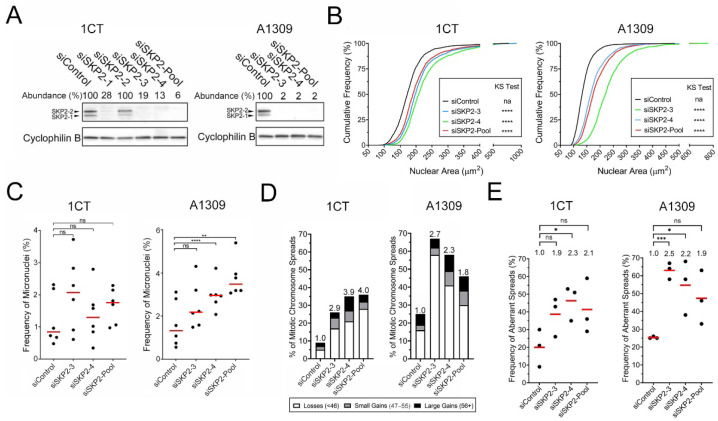
*SKP2* silencing in 1CT and A1309 cells corresponds with increases in CIN-associated phenotypes. (**A**) Semi-quantitative Western blots depicting reduced SKP2 abundance following silencing relative to siControl in 1CT (left) and A1309 (right) cells. (**B**) Cumulative distribution frequency graphs reveal statistically significant increases in nuclear areas following *SKP2* silencing (two-sample KS tests; na, not applicable; ****, *p*-value < 0.0001; N = 3, n = 6). (**C**) Dot plots depicting trending (1CT) and significant (A1309) increases in the frequency of micronuclei following *SKP2* silencing relative to siControl (MW test; ns, not significant, *p*-value > 0.05; **, *p*-value < 0.01; ****, *p*-value < 0.0001; N = 3, *n* = 6) (**D**) Bar graphs depicting the frequency of spreads with aberrant chromosome numbers following *SKP2* silencing. Fold increase relative to siControl is presented at the top of each column (N = 3, n = 100 spreads/condition). (**E**) Dot plots reveal significant increases in the frequency of total aberrant chromosome spreads following *SKP2* silencing relative to siControl; fold increase relative to siControl is indicated. (Student’s *t*-test; ns, not significant, *p*-value > 0.05; *, *p*-value ≤ 0.05; ***, *p*-value < 0.001; N = 3, n = 100/condition).

**Figure 6 cells-11-03731-f006:**
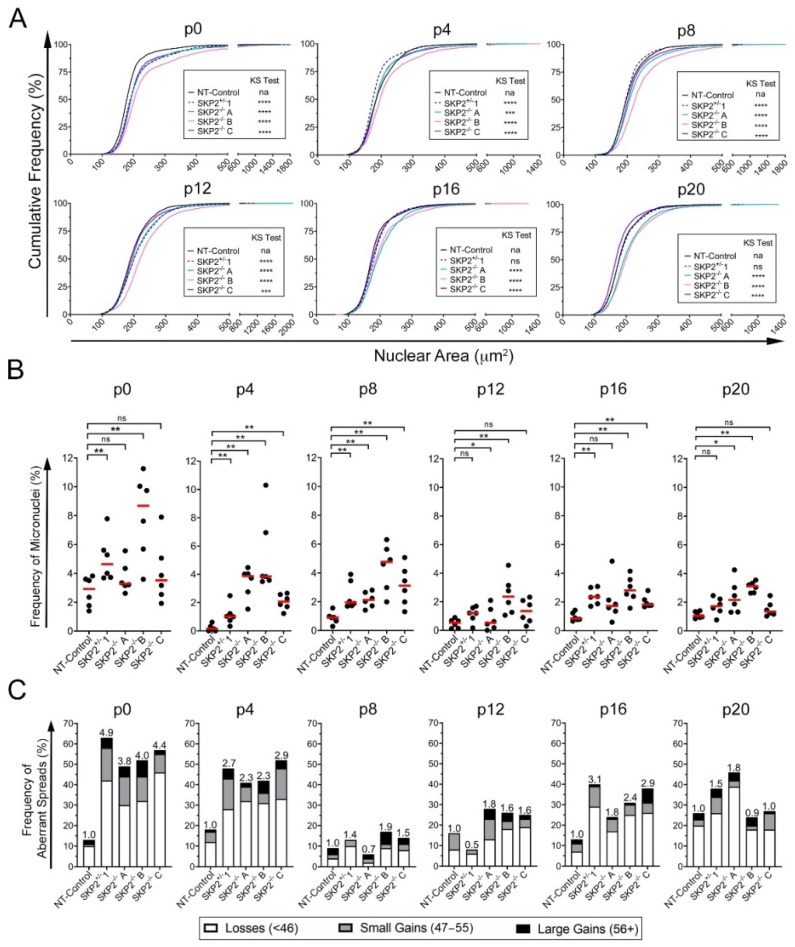
*SKP2* Knockout Clones Display Dynamic Changes in CIN Phenotypes. (**A**) Cumulative distribution graphs reveal dynamic and significant changes in nuclear areas within *SKP2*^+/−^ and *SKP2*^−/−^ clones over 10 weeks (p0 to p20) relative to NT-Control (two-sample KS test; na, not applicable; ***, *p*-value < 0.001; ****, *p*-value < 0.0001; N = 1; n = > 1000 nuclei/condition). (**B**) Dot plots uncover significant and dynamic increases in the frequency of micronuclei within *SKP2*^+/−^ and *SKP2*^−/−^ clones from p0 to p20 relative to NT-Control (MW test; ns, not significant, *p*-value > 0.05; *, *p*-value ≤ 0.05; **, *p*-value < 0.01; N = 1; n = 6). (**C**) Bar graph depicting the frequencies of spreads with aberrant chromosome numbers, including chromosome losses, small-scale gains, and large-scale gains in *SKP2*^+/−^ and *SKP2*^−/−^ clones over 10 weeks (p0 to p20) relative to NT-Control. The fold increase in total frequencies of aberrant spreads relative to NT-Control is presented above each bar (N = 1, n = 100 spreads/condition).

**Figure 7 cells-11-03731-f007:**
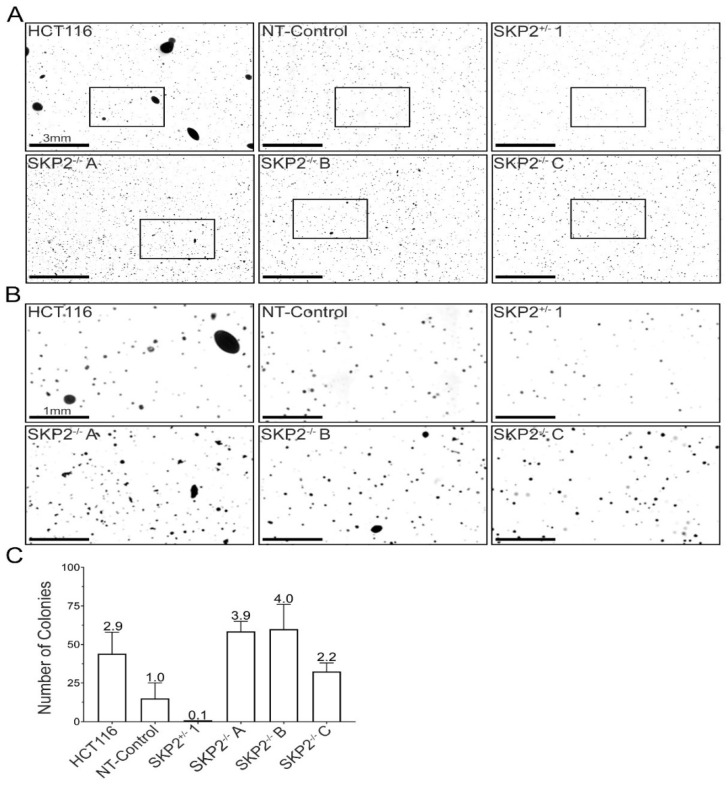
Reduced *SKP2* Expression Alters Anchorage-Independent Growth. (**A**) Low-resolution montages presenting colony formation in soft agar. Bounding boxes identify the corresponding magnified regions presented in (**B**). (**C**) Bar graph presenting the mean number of colonies (≥100 µm in diameter), with the mean fold increases relative to NT-Control presented above each column. (N = 1, n = 2).

## Data Availability

Patient-related data (Figure 2) are based upon data generated by the TCGA Research Network and are available at https://www.cancer.gov/tcga, accessed on 15 June 2022. All descriptive statistics and statistical analyses presented in Figure 3, Figure 4, Figure 5, Figure 6 and Figure 7 are provided in Appendix A.

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
