# Peer review of "Reduced SKP2 Expression Adversely Impacts Genome Stability and Promotes Cellular Transformation in Colonic Epithelial Cells"

_cells, 2022, doi:10.3390/cells11233731_

Round 1

Reviewer 1 Report

Neudorf et al present a manuscript describing the role of SKP2 in genomic stability and in  promoting transformation in colonic epithelial cells. The authors used siRNA based screen to identify F-box genes that affect overall nuclear area and size and found that SKP2 leads to a significant increase in the nuclear area suggesting that it could lead to chromosomal instability. They further go on to validate this phenotype by directly looking at chromosomal metaphase spreads and micronuclei. They also generate independent CRISPR-Cas9 mediated knockout clones and determine changes in CIN phenotype. Overall, this is a sound study and some of the conclusions are well thought out. However, many conclusions are not supported by appropriate evidence. I suggest that authors show some additional evidence of CIN to support their claims. This manuscript needs to be revised before it can be accepted for publication. 

1. The authors claim that micronuclei are hallmarks of CIN. However, in Figure 3D and 3E, they do not observe any changes in the micronuclei after SKP2 knockdown. This is running counter to their claim and suggests that SKP2 knockdown does not lead to CIN. 

2. In Fig. 3A, the 2 bands are marked as SKP2-1 and SKP2-2. The siRNAs are also marked by the same name where SKP2-1 leads to disappearance of both bands while SKP2-2 does not. What exactly are the 2 bands? Are the labelings accurate?

3. Fig 4A. These images do not look at the same magnification. Please provide a scale-bar. It could be possible that higher magnification may have left out some chromosomes that are not exactly very close to the spreads. 

4. In Fig. 6A, the figures do not support the text very well. There does not appear to be any significant observable difference in the nuclear area between p0 to p20. This evidence is not strong enough and needs to be supported further. 

5. Fig 6B, Why does the frequency of micronuclei overall decrease with time and passages? If anything, this result suggests that these cells naturally have a high frequency of micronuclei and after removing SKP2 those differences go away. Similarly, in Fig 6C, the differences in aberrant spreads go away after passaging several fold. 

6. Page 6, line 268,269: "..reduced SKP2 expression may also be a CIN gene." Please modify this text. SKP2 is a gene involved in CIN or something along the lines. Reduced expression cannot be a gene.   

7.  Page 12, Subsection 3.7: Remove the word subsection from the text. 

Author Response

       We thank the Reviewer for reading our manuscript and providing constructive critique and suggestions to enhance our manuscript. We believe the current iteration represents a significant improvement over the original submission and hope that you will accept it for publication in Cells. Below are our point-by-point responses to each of the Reviewer’s concerns/suggestions:

Neudorf et al present a manuscript describing the role of SKP2 in genomic stability and in promoting transformation in colonic epithelial cells. The authors used siRNA-based screen to identify F-box genes that affect overall nuclear area and size and found that SKP2 leads to a significant increase in the nuclear area suggesting that it could lead to chromosomal instability. They further go on to validate this phenotype by directly looking at chromosomal metaphase spreads and micronuclei. They also generate independent CRISPR-Cas9 mediated knockout clones and determine changes in CIN phenotype. Overall, this is a sound study and some of the conclusions are well thought out. However, many conclusions are not supported by appropriate evidence. I suggest that authors show some additional evidence of CIN to support their claims. This manuscript needs to be revised before it can be accepted for publication.

  1. The authors claim that micronuclei are hallmarks of CIN. However, in Figure 3D and 3E, they do not observe any changes in the micronuclei after SKP2 knockdown. This is running counter to their claim and suggests that SKP2 knockdown does not lead to CIN.

       We wish to highlight that while micronuclei are hallmarks or CIN, their presence alone is not mandatory to identify cells exhibiting a CIN phenotype. That is, not all cells exhibiting CIN will exhibit increases in micronucleus formation. Importantly, the underlying mechanism(s) leading to CIN are diverse and those underlying micronucleus formation are typically associated with defects in DNA double strand break repair and the formation of acentric chromosomal fragments, which cannot be properly segregated during mitosis (anaphase to telophase) due to the lack of kinetochore attachments. Thus, it is the biological differences in the underlying mechanisms giving rise to CIN that mandate multiple, complementary CIN assays be performed and is why we purposefully perform three complementary, yet distinct CIN-based assays, namely nuclear areas, micronucleus formation and chromosome enumeration. To enhance clarity, we now highlight the importance of employing a multi-pronged approach to identify CIN genes and have included the following statement (Page 16, Lines 625-28); “It should be noted that changes in nuclear areas and increased micronucleus formation typically arise through aberrations in distinct cellular pathways including cell cycle dysregulation or centrosome overduplication, and mitotic spindle dynamics or errors in double-strand break repair, respectively[1,2]”. Thus, both assays are independent surrogate markers for CIN and are suggestive of the mechanism by which a putative CIN gene may be inducing increased CIN. Further, the large increases in nuclear areas observed following reduced SKP2 expression and evidence that SKP2 targets Cyclin E1 for proteolytic degradation we observe in the current study coupled with several additional studies investing the core SCF complex members[3-6] strongly support the possibility that reduced SKP2 expression induces CIN through cell cycle defects including endoreduplication.

  1. In Fig. 3A, the 2 bands are marked as SKP2-1 and SKP2-2. The siRNAs are also marked by the same name where SKP2-1 leads to disappearance of both bands while SKP2-2 does not. What exactly are the 2 bands? Are the labelings accurate?

       The two bands present within the western blot are the two established and expressed isoforms of SKP2 [7,8]. The four SKP2 siRNA duplexes predicted to silenced SKP2 expression were identified by Dharmacon using their proprietary software algorithms. As the secondary and tertiary structure of the mRNA can render the corresponding regions inaccessible to individual siRNA duplexes, some duplexes may be more effective than others. This is precisely why we test all four duplexes to identify those with the greatest silencing efficiencies. As is the case for SKP2, siSKP2-3 and -4 were identified as the two most effective siRNA duplexes.

       We previously distinguished the two protein isoforms from the siRNA duplexes in Figure 3A, we labeled the isoforms as SKP2-1 and SKP2-2, and the duplexes as siSKP2-1, -2, -3, -4 and -Pool. However, to enhance clarity, we have amended the figure legend to include (Page 8; Lines 311-314): “Western blots presenting SKP2 abundance following siRNA-based silencing with individual (siSKP2-1, -2, -3, -4; top labels) and pooled (siSKP2-Pool) siRNAs relative to siControl; Cyclophilin B serves as the loading control. Note that two bands representing the two SKP2 isoforms (SKP2-1 and SKP2-2; left labels) are visible”.

  1. Fig 4A. These images do not look at the same magnification. Please provide a scale-bar. It could be possible that higher magnification may have left out some chromosomes that are not exactly very close to the spreads.

       We highlight that the images are originally collected using a 63× objective such that the entirety of each mitotic chromosome spread is clearly contained and entirely visible. The representative images presented in Figure 4A are cropped portions of those images and are simply used to provide examples of the various categories including near diploid, losses, small gains and large gains. Moreover, the original panels were purposefully presented to provide maximal image information for each individual spread to best present the images for the reader. As the goal of this work is simply to enumerate chromosomes, the sizes of the individual chromosomes was not deemed relevant and was the reason why scale bars were not included. Nevertheless, we have now amended Figure 4A so that each panel is presented at a slightly lower magnification (increased black space around each spread) and a scale bar (10mm) has also been included.

  1. In Fig. 6A, the figures do not support the text very well. There does not appear to be any significant observable difference in the nuclear area between p0 to p20. This evidence is not strong enough and needs to be supported further.

       While the Reviewer indicates there is no “significant observable difference in the nuclear area between p0 to p20”, nearly all statistical comparisons for the cumulative nuclear area distribution frequencies for each clone and the respective NT-Control at each passage are statistically significant and distinct. So while the curves may visually appear similar, the statistical tests (two sample Kolmogorov-Smirnov tests) regularly identify statistically significant differences in their distributions. Moreover, the clones exhibit dynamic changes in nuclear area distributions throughout the experiment (i.e., leftward and rightward shifts) with most exhibiting p-values <0.0001, which is below the conventional threshold of p <0.05, and that stated within the Material and Methods of the current study (data were also provided as Supplementary Table S9). As nuclear areas are reflective of the quantity of DNA contained within the nucleus, these results indicate progressive large-scale changes in the DNA content of the nuclei of SKP2 clones across time[9].

       The Reviewer also indicates that the findings of the cumulative nuclear area distribution frequencies is not strong and needs to be supported further. While we disagree with this statement as we demonstrated statistical significance (see preceding paragraph), we have already provided additional evidence and support through two additional assays – micronucleus formation and chromosome enumeration. These data were provided in the original submission as Figures 6B and 6C, respectively, and remain in the current iteration. Accordingly, it is our belief that the data gleaned from the three complementary CIN assays, provide compelling evidence that reduced SKP2 expression induces CIN in a non-malignant/non-transformed cellular context (A1309).

  1. Fig 6B, Why does the frequency of micronuclei overall decrease with time and passages? If anything, this result suggests that these cells naturally have a high frequency of micronuclei and after removing SKP2 those differences go away. Similarly, in Fig 6C, the differences in aberrant spreads go away after passaging several-fold.

       We agree with the Reviewer that the frequency of micronucleus formation generally declines with time; however, we disagree with the comment that “this result suggests that these cells naturally have a high frequency of micronuclei and after removing SKP2 those difference go away”. We highlight that A1309 cells have a low frequency of micronuclei as presented in several different contexts. For example, Figure 5C shows that A1309 cells have a mean frequency of ~1%, while Figure 6B shows that micronucleus formation of the NT-Control clone is similar with a 1-3% mean frequency. In fact, our original Discussion addressed this very issue and stated (Page 16; Lines 621-3); “Importantly, it should be re-iterated that the KRAS, APC and TP53 alterations by themselves, do not induce micronucleus formation, as the frequency of micronuclei is similar and low (~1%) in both 1CT and A1309”, which remains in the current submission (Page 16; Lines 628-30). Furthermore, micronucleus formation within the SKP2 clones is significantly higher (*, p< 0.05; **, p<0.01) than the NT-Control in each time point of the experiment and exhibit dynamic changes between passages as indicated in the statistical information presented above each dot plot.

       The underlying biological reason accounting for the decreasing frequency of micronuclei over time remains unknown and is beyond the scope of the current manuscript, but we theorize most likely has something to do with the aberrant pathways arising due to aberrant SCF formation and function. For example, the increased micronucleus formation that arises following decreased SKP2 expression is suggestive of extensive DNA double-strand breaks[10,11]. Thus, the extensive micronucleus formation observed within the first few passages is likely incompatible with cell viability and cells producing large amounts of micronuclei are selected against over time (i.e., die and are lost from the population).

       We also agree with the Reviewer’s observation that SKP2 clones exhibit increased frequencies of aberrant chromosome spreads in early passages that decrease over time; however, we again highlight that these aberrant karyotypes remain increased within the SKP2 clones relative to the NT-Control clone at each timepoint and agree with the data presented in Figure 5E, which show that cells with reduced SKP2 expression have significant increases in abnormal karyotypes across three biological replicates. Like the data presented in Figure 6A and 6B, chromosome complements are dynamic and heterogenous across time points for the SKP2 clones relative to NT-Control and provide further evidence that reduced SKP2 expression induces CIN. Finally, we would like to emphasize that it is indeed cellular populations exhibiting low to intermediate CIN levels that are proposed to be the predominant drivers of disease development and progression[12-17], supporting the possibility that SKP2 is a novel CIN gene with potential implications for disease development. 

  1. Page 6, line 268,269: "..reduced SKP2 expression may also be a CIN gene." Please modify this text. SKP2 is a gene involved in CIN or something along the lines. Reduced expression cannot be a gene.

       For enhanced clarity, we have amended the sentence as follows (Page 7; Lines 278-80): “…support the possibility that reduced SKP2 expression may also induce CIN”.

  1. Page 12, Subsection 3.7: Remove the word subsection from the text.

       Corrected as suggested by the Reviewer.

REFERENCES:

  1. Fenech, M. Chromosomal biomarkers of genomic instability relevant to cancer. Drug Discov Today 2002, 7, 1128-1137, doi:10.1016/s1359-6446(02)02502-3.
  2. Markossian, S.; Arnaoutov, A.; Saba, N.S.; Larionov, V.; Dasso, M. Quantitative assessment of chromosome instability induced through chemical disruption of mitotic progression. Cell Cycle 2016, 15, 1706-1714, doi:10.1080/15384101.2016.1175796.
  3. Nakayama, K.; Nagahama, H.; Minamishima, Y.A.; Matsumoto, M.; Nakamichi, I.; Kitagawa, K.; Shirane, M.; Tsunematsu, R.; Tsukiyama, T.; Ishida, N.; et al. Targeted disruption of Skp2 results in accumulation of cyclin E and p27(Kip1), polyploidy and centrosome overduplication. Embo j 2000, 19, 2069-2081, doi:10.1093/emboj/19.9.2069.
  4. Nakayama, K.; Nagahama, H.; Minamishima, Y.A.; Miyake, S.; Ishida, N.; Hatakeyama, S.; Kitagawa, M.; Iemura, S.; Natsume, T.; Nakayama, K.I. Skp2-mediated degradation of p27 regulates progression into mitosis. Dev Cell 2004, 6, 661-672, doi:10.1016/s1534-5807(04)00131-5.
  5. Nakayama, K.I.; Hatakeyama, S.; Nakayama, K. Regulation of the cell cycle at the G1-S transition by proteolysis of cyclin E and p27Kip1. 2001, 282, 853-860, doi:10.1006/bbrc.2001.4627.
  6. Thompson, L.L.; Baergen, A.K.; Lichtensztejn, Z.; McManus, K.J. Reduced SKP1 expression induces chromosome instability through aberrant cyclin E1 protein turnover. Cancers 2020, 12, 531-531, doi:10.3390/cancers12030531.
  7. The Human Protein Atlas. SKP2: Protein Information. Available online: https://www.proteinatlas.org/ENSG00000145604-SKP2 (accessed on April 1, 2022)
  8. UniprotKB. SKP2: Family & Domains. Available online: https://www.uniprot.org/uniprot/Q13309 (accessed on April 1).
  9. Geigl, J.B.; Obenauf, A.C.; Schwarzbraun, T.; Speicher, M.R. Defining 'chromosomal instability'. Trends in Genetics 2008, 24, 64-69, doi:10.1016/j.tig.2007.11.006.
  10. Bhatia, A.; Kumar, Y. Cancer cell micronucleus: an update on clinical and diagnostic applications. APMIS 2013, 121, 569-581, doi:10.1111/apm.12033.
  11. Stopper, H.M., S. O. Micronuclei as a biological endpoint for genotoxicity: A minireview. Toxicol. Vitr. 1997, 11, 661–667.
  12. Janssen, A.; Kops, G.J.; Medema, R.H. Elevating the frequency of chromosome mis-segregation as a strategy to kill tumor cells. Proc Natl Acad Sci U S A 2009, 106, 19108-19113, doi:10.1073/pnas.0904343106.
  13. Jeusset, L.; McManus, K. Developing Targeted Therapies That Exploit Aberrant Histone Ubiquitination in Cancer. Cells 2019, 8, 165-165, doi:10.3390/cells8020165.
  14. Bakhoum, S.F.; Compton, D.A. Chromosomal instability and cancer: A complex relationship with therapeutic potential. Journal of Clinical Investigation 2012, doi:10.1172/JCI59954.
  15. Sajesh, B.V.; Guppy, B.J.; McManus, K.J. Synthetic genetic targeting of genome instability in cancer. Cancers (Basel) 2013, 5, 739-761, doi:10.3390/cancers5030739.
  16. Sajesh, B.V., Bailey, M., Lichtensztejn, Z., Hieter, P. & McManus, K. J. . Synthetic lethal targeting of superoxide dismutase 1 selectively kills RAD54B-deficient colorectal cancer cells. Genetics 2013, 195, 757–767.
  17. Guppy, B.J.; McManus, K.J. Synthetic lethal targeting of RNF20 through PARP1 silencing and inhibition. Cell Oncol (Dordr) 2017, 40, 281-292, doi:10.1007/s13402-017-0323-y.

Reviewer 2 Report

This manuscript describes the impact of reduced SKP2 expression, a F-box gene in the SCF complex, on genomic stability of colonic epithelial cells and progression of colorectal carcinomas (CRC). Based on measurements on cell lines, chiefly by quantitative imaging microscopy (QIM), the authors proposed that SKP2 is a novel chromosomal instability (CIN) gene. While the results are interesting, there are concerns on the interpretations and conclusions.

Major comments for authors:

1.      Experiments are performed on one malignant CRC cell line HCT116 only. As the authors pointed out, this is a microsatellite unstable, KRAS mutated cell line. Microsatellite high human CRC tumors develop by mismatch repair/DNA defect, not by CIN. Hence, this is not an appropriate cell line to study the effect of reduced SKP2 on CIN. The experiments should be repeated on at least two microsatellite stable (MSS) cell lines.  In this regard, the authors may wish to choose cell lines that mimic the human CRC tumor tissues as much as possible (see e.g., Ronen et al., Life Science Alliance 2019).

2.      The clinicopathological features of the TCGA data (section 3.2) should be described in more details e.g., are they MSS tumors? In addition, the authors described SKP2 copy number loss is associated with worse progression in CRC. How about SKP2 copy number gain or mutations?

3.      The authors discussed the controversy of whether SKP2 is oncogenic or tumor suppressive. If so, the authors should also measure the effect of reduced SKP2 expression on cyclin E1 and p27 levels in the experiments using the various SKP2 constructs (sections 3.3-7).

4.      Further, the experiments using CRISPR-cas 9 mutated SKP2 alleles in A1309 cell line (section 3.6 and Figure 6) have variable results (e.g., they do not generate more micronuclei after passage 8). This seems to suggest that the constructs are not stable. Why not perform these experiments in an appropriate malignant cell line (see comment 1 above)?

5.      The authors should consider a second assay e.g., measurement of aneuploidy to substantiate the results obtained by QIM.

Minor Comments:

1.      In Fig. 1, the results can be better represented (smaller range on Y-axis) if the outlier point in FBXL7 is removed.

2.      In Figures 3A and 5A, the unit (%) should be indicated on the headings for the lanes to enable better comprehension of the Western blots.

Author Response

We thank the Reviewer for reading our manuscript and providing constructive critique and suggestions to enhance our manuscript. We believe the current iteration represents a significant improvement over the original submission and hope that you will accept it for publication in Cells. Below are our point-by-point responses to each of the Reviewer’s concerns/suggestions:

This manuscript describes the impact of reduced SKP2 expression, a F-box gene in the SCF complex, on genomic stability of colonic epithelial cells and progression of colorectal carcinomas (CRC). Based on measurements on cell lines, chiefly by quantitative imaging microscopy (QIM), the authors proposed that SKP2 is a novel chromosomal instability (CIN) gene. While the results are interesting, there are concerns on the interpretations and conclusions.

Major comments for authors:

  1. Experiments are performed on one malignant CRC cell line HCT116 only. As the authors pointed out, this is a microsatellite unstable, KRAS mutated cell line. Microsatellite high human CRC tumors develop by mismatch repair/DNA defect, not by CIN. Hence, this is not an appropriate cell line to study the effect of reduced SKP2 on CIN. The experiments should be repeated on at least two microsatellite stable (MSS) cell lines. In this regard, the authors may wish to choose cell lines that mimic the human CRC tumor tissues as much as possible (see e.g., Ronen et al., Life Science Alliance 2019).

       We respectfully disagree with the Reviewer that the experiments are only conducted in HCT116 and that this is an inappropriate line to use as it exhibits MSI. First, a total of three colonic epithelial cell lines were employed including HCT116, 1CT and A1309. These cells, and HCT116 in particular, were purposefully selected as they are all karyotypically stable (HCT116 has a modal chromosome number = 45), which is mandatory to study genes inducing CIN. That is, one cannot study CIN in a cell line already exhibiting CIN and so HCT116, 1CT and A1309 are ideal cell models in which to study CIN. Additionally, all three cell lines, and HCT116 in particular have been previously vetted by numerous reviewers and used in an extensive array of CIN-based studies[1-7]. We also wish to highlight that as CIN is associated with 85% of all CRCs, most CRC cell lines naturally exhibit CIN and therefore are inappropriate for this study. Finally, we agree with the Reviewer that HCT116 are an MSI line and freely acknowledged this within the original Discussion, which stated (Page 16; Lines 603-7), “… HCT116 is a microsatellite instability cell line that contains a MutL Homolog 1 (MLH1) deficiency underlying defects in DNA mis-match repair[76]. Thus, these cells harbor a mutator phenotype that may produce mutations capable of synergizing with reduced SKP2 expression to enhance the various CIN phenotypes assessed” and this statement remains within the current submission (Page 16; Lines 610-614). Thus, in response to the Reviewer’s statement, we have provided empirical data that support our conclusion that reduced SKP2 expression induces CIN in three appropriate and karyotypically stable, colonic epithelial cell contexts.

  1. The clinicopathological features of the TCGA data (section 3.2) should be described in more details e.g., are they MSS tumors? In addition, the authors described SKP2 copy number loss is associated with worse progression in CRC. How about SKP2 copy number gain or mutations?

       We thank the Reviewer for their insightful query and in agreement with reduced SKP2 expression inducing CIN, further review of the TCGA Pan-Cancer Atlas data revealed that 98.4% (63/64 cases) of patient samples with SKP2 copy number losses exhibit CIN, while the remaining case was MSI positive. Based on these new findings, we have amended the manuscript to include the following (Page 6; Lines 258-60): “Subsequent analyses of CRC patient tumor samples with SKP2 copy number losses revealed that 98.4% (63/64 cases) exhibit CIN, further supporting the possibility that reduced SKP2 expression induces CIN”.

       Although not the focus of the current study, similar health outcome analyses were performed for CRC patient samples with copy number gains and mutations. No statistical differences (p-values >0.05) in survival outcomes occurred when comparing cases with SKP2 copy number gains or SKP2 mutations to diploid cases. Thus, based on the analyses, only SKP2 copy number losses are associated with statistically worse outcomes. This information remains unchanged within the current submission.

  1. The authors discussed the controversy of whether SKP2 is oncogenic or tumor suppressive. If so, the authors should also measure the effect of reduced SKP2 expression on cyclin E1 and p27 levels in the experiments using the various SKP2 constructs (Sections 3.3-7).

       As suggested by the Reviewer, we have included Western blots evaluating the abundance of P27, a representative example of a substrate protein normally targeted by the SCFSKP2 complex for proteolytic degradation by the 26S proteasome. Within the Supplementary Information, we now provide additional figures demonstrating increases in P27 abundance following both silencing of P27 in HCT116 (Figure S2) and A1309 (Figure S4) cells and following CRISPR/Cas9 knockout within the A1309 clones (Figure S7). Additionally, we have incorporated additional text within the body of the manuscript to reflect this additional information. More specifically, we have added the following text (Page 7; Lines 286-8); “Moreover, western blotting also revealed that SKP2 silencing corresponded with increases in P27 abundance indicating that reduced SKP2 expression adversely impacted normal SCFSKP2 function and proteolytic targeting of P27 (Figure S2)”. We also amended a sentence to more clearly indicate aberrant targeting, which now reads as (Page 9; Lines 378-82); “Prior to performing the CIN assays, western blots were performed and determined that the silencing efficiencies of the individual (siSKP2-3 and -4) and pool (siSKP2-Pool) siRNAs were highly effective within both lines and generally reduced expression to <10% of siControl levels (Figure 5A; Figure S3), which corresponded with P27 accumulation in A1309 cells (Figure S4)”. Finally, with respect to the CRISPR/Cas9 clones, we included the following statement (Page 11; Lines 436-8); “Western blotting also revealed that the SKP2 clones exhibited increases in P27 abundance relative to NT-Control (Figure S7)”.

  1. Further, the experiments using CRISPR-cas9 mutated SKP2 alleles in A1309 cell line (Section 3.6 and Figure 6) have variable results (e.g., they do not generate more micronuclei after passage 8). This seems to suggest that the constructs are not stable. Why not perform these experiments in an appropriate malignant cell line (see comment 1 above)?

       While we do not disagree with the Reviewer that the CRISPR-cas9 mutated SKP2 alleles have variable results, we do disagree with their statement that they do not generate more micronuclei after passage 8. As shown in Figure 6B, 6/9 pairwise statistical comparisons from p12-20 reveal significant increases (*, <0.05; **, <0.01) relative to the NT-Control, while those that are not significant tend to show increasing trends relative to the NT-Control. In any case, these variable and ever-changing phenotypes are expected in the context of CIN, as this is precisely what the CIN phenotype induces – ongoing changes – some of which may be selected for, as they may provide selective advantages, and some of which may be selected against, as they confer growth/survival disadvantages. For example, the increased micronucleus formation that arises following decreased SKP2 expression in the earlier passages is suggestive of extensive DNA double-strand breaks[8,9]. Thus, the extensive micronucleus formation observed within the first few passages is likely incompatible with cell viability and cells producing large amounts of micronuclei are selected against over time. Similarly, the SKP2 clones exhibit high frequencies of abnormal chromosome numbers within the first passages that also decline throughout the experiment. However, we emphasize that these aberrant karyotypes remain elevated relative to the NT-Control clone in each passage. These data are further supported by those presented in Figure 5E, which demonstrate that cells with reduced SKP2 expression have significant increases in abnormal karyotypes across three biological replicates that include 100 nuclei analyzed per experimental condition. As observed within the data presented in Figure 6A and 6B, chromosome complements within the clones are dynamic and heterogenous across time, providing strong evidence that reduced SKP2 expression induces CIN. We also highlight that it is indeed cellular populations exhibiting low to intermediate CIN levels that are proposed to be the predominant drivers of disease development and progression[10-15], supporting the possibility that SKP2 is a novel CIN gene with potential implications for disease development.

       The distinct CIN phenotypes observed between SKP2 clones during the time course experiment suggest they underwent distinct karyotypic evolutions throughout the clonal expansion process. These clones arose from a single cell deficient of a suspected CIN gene (i.e., homozygous/heterozygous loss of SKP2), therefore, an increased rate of chromosome gains and losses is expected to occur during clonal expansion, leading to karyotypic differences within and between each clone. Thus, it is most likely that the four clones (SKP2+/-1, SKP2-/-A, SKP2-/-B and SKP2-/-C) evolved differently depending on the chromosome complements present during the early stages of clonal expansion and whether certain karyotypes conferred selective growth and survival advantages. The heterogeneity observed between the SKP2 clones employed within this study agrees with that observed in other studies that have also assessed CIN genes over time[6,16]. For example, a similar study investigating the impact of reduced FBXO7 expression on CIN found that although two distinct homozygous clones exhibited a similar reduction in protein abundance (~0%), the CIN phenotypes associated with each individual clone were notably distinct despite arising from syngeneic backgrounds[17]. We also confirm that the SKP2 “constructs” or gene knockout clones are stable with respect to SKP2 loss over time as the western blot presented in Figure S5 and S6 show the expected SKP2 expression levels after 20 passages.

       With respect to the suggestion of conducting these experiments in an appropriate malignant cell line. As the central focus of this study was to determine the impact reduced SKP2 expression has on early disease development (i.e., cellular transformation), only non-transformed/non-malignant cells can be employed. As described above (see Comment #1), although HCT116 are an ideal model to study CIN as they are karyotypically stable, they were not selected for this aspect of the study as they harbor a mutator phenotype, which over the extended time course of this study would likely adversely impact the interpretation of the results. Perhaps even more important, HCT116 (or any cancer cell line) are transformed and therefore are incapable of providing any meaningful insight into the impact reduced SKP2 expression has on early disease development (i.e., cellular transformation). Thus, the A1309 cells are an ideal non-transformed cellular context in which to study the impact on early disease development such as cellular transformation (see Section 3.7).

  1. The authors should consider a second assay e.g., measurement of aneuploidy to substantiate the results obtained by QIM.

       We agree that multiple assays should be performed to comprehensively assess the CIN phenotype and is precisely why routinely performed three complementary assays to assess CIN including nuclear areas, micronucleus formation and chromosome enumeration. We believe that our three CIN assays provide reproducible and strong evidence that reduced SKP2 expression induces CIN phenotypes and changes in chromosome complements (i.e., aneuploidy). In essence, the mitotic chromosome spreads/chromosome enumeration assays do assess aneuploidy by simply counting the number of chromosomes within a given spread. We did not perform karyotypic analyses to assign specific karyotypes to each of the various conditions, as this is extremely laborious, time consuming and costly (Spectral Karyotyping; SKY) and is unlikely to identify specific clones, as CIN drives ever changing chromosome complements over time (see Figure 6). Moreover, it is our experience, both with patient samples[18-21] and cell lines[2,6,22,23], that CIN drives ongoing changes in chromosome complements and that the gain or loss of specific chromosomes appears largely random. Nevertheless, we did generate and enumerate mitotic chromosome spreads for each cell line and condition (Figure 4B and C; Figure 5D and E; Figure 6C) that clearly show that cells with reduced SKP2 expression exhibit significant increases in aberrant chromosome complements (i.e., aneuploidy). Thus, our assays inherently assess the frequency of aberrant chromosome spreads and aneuploidy, which collectively support our conclusion that reduced SKP2 expression induces CIN in various colonic epithelial cell contexts.

Minor Comments:

  1. In Fig. 1, the results can be better represented (smaller range on Y-axis) if the outlier point in FBXL7 is removed.

We have amended Figure 1 as suggested (y-axis range has been decreased).

  1. In Figures 3A and 5A, the unit (%) should be indicated on the headings for the lanes to enable better comprehension of the Western blots.

       To enhance clarity, we have amended the Figures 3A, 5A, S1, S3 and S6, and have now included “Abundance (%)” next to the numerical values in each of the western blots.

REFERENCES:

  1. Thompson, L.L.; McManus, K.J. A novel multiplexed, image-based approach to detect phenotypes that underlie chromosome instability in human cells. PLoS One 2015, 10, e0123200, doi:10.1371/journal.pone.0123200.
  2. Thompson, L.L.; Baergen, A.K.; Lichtensztejn, Z.; McManus, K.J. Reduced SKP1 expression induces chromosome instability through aberrant cyclin E1 protein turnover. Cancers 2020, 12, 531-531, doi:10.3390/cancers12030531.
  3. Asbaghi, Y.; Thompson, L.L.; Lichtensztejn, Z.; McManus, K.J. KIF11 silencing and inhibition induces chromosome instability that may contribute to cancer. Genes Chromosomes Cancer 2017, 56, 668-680, doi:10.1002/gcc.22471.
  4. Guppy, B.J.; McManus, K.J. Mitotic accumulation of dimethylated lysine 79 of histone H3 is important for maintaining genome integrity during mitosis in human cells. Genetics 2015, 199, 423-433, doi:10.1534/genetics.114.172874.
  5. Barber, T.D.; McManus, K.; Yuen, K.W.; Reis, M.; Parmigiani, G.; Shen, D.; Barrett, I.; Nouhi, Y.; Spencer, F.; Markowitz, S.; et al. Chromatid cohesion defects may underlie chromosome instability in human colorectal cancers. Proc Natl Acad Sci U S A 2008, 105, 3443-3448, doi:10.1073/pnas.0712384105.
  6. Palmer, M.C.L.; Neudorf, N.M.; Farrell, A.C.; Razi, T.; Lichtensztejn, Z.; McManus, K.J. The F-box protein, FBXO7 is required to maintain chromosome stability in humans. Hum Mol Genet 2021, doi:10.1093/hmg/ddab330.
  7. Thompson, L. Microscopy-based High-content Analyses Identify Novel Chromosome Instability Genes including SKP1. 2018.
  8. Bhatia, A.; Kumar, Y. Cancer cell micronucleus: an update on clinical and diagnostic applications. APMIS 2013, 121, 569-581, doi:10.1111/apm.12033.
  9. Stopper, H.M., S. O. Micronuclei as a biological endpoint for genotoxicity: A minireview. Toxicol. Vitr. 1997, 11, 661–667.
  10. Janssen, A.; Kops, G.J.; Medema, R.H. Elevating the frequency of chromosome mis-segregation as a strategy to kill tumor cells. Proc Natl Acad Sci U S A 2009, 106, 19108-19113, doi:10.1073/pnas.0904343106.
  11. Jeusset, L.; McManus, K. Developing Targeted Therapies That Exploit Aberrant Histone Ubiquitination in Cancer. Cells 2019, 8, 165-165, doi:10.3390/cells8020165.
  12. Bakhoum, S.F.; Compton, D.A. Chromosomal instability and cancer: A complex relationship with therapeutic potential. Journal of Clinical Investigation 2012, doi:10.1172/JCI59954.
  13. Sajesh, B.V.; Guppy, B.J.; McManus, K.J. Synthetic genetic targeting of genome instability in cancer. Cancers (Basel) 2013, 5, 739-761, doi:10.3390/cancers5030739.
  14. Sajesh, B.V., Bailey, M., Lichtensztejn, Z., Hieter, P. & McManus, K. J. . Synthetic lethal targeting of superoxide dismutase 1 selectively kills RAD54B-deficient colorectal cancer cells. Genetics 2013, 195, 757–767.
  15. Guppy, B.J.; McManus, K.J. Synthetic lethal targeting of RNF20 through PARP1 silencing and inhibition. Cell Oncol (Dordr) 2017, 40, 281-292, doi:10.1007/s13402-017-0323-y.
  16. Jeusset, L.M.; Guppy, B.J.; Lichtensztejn, Z.; McDonald, D.; McManus, K.J. Reduced USP22 Expression Impairs Mitotic Removal of H2B Monoubiquitination, Alters Chromatin Compaction and Induces Chromosome Instability That May Promote Oncogenesis. Cancers (Basel) 2021, 13, doi:10.3390/cancers13051043.
  17. Petersen, I.; Kotb, W.F.; Friedrich, K.H.; Schluns, K.; Bocking, A.; Dietel, M. Core classification of lung cancer: correlating nuclear size and mitoses with ploidy and clinicopathological parameters. Lung Cancer 2009, 65, 312-318, doi:10.1016/j.lungcan.2008.12.013.
  18. Morden, C.R.; Farrell, A.C.; Sliwowski, M.; Lichtensztejn, Z.; Altman, A.D.; Nachtigal, M.W.; McManus, K.J. Chromosome instability is prevalent and dynamic in high-grade serous ovarian cancer patient samples. Gynecol Oncol 2021, 161, 769-778, doi:10.1016/j.ygyno.2021.02.038.
  19. Penner-Goeke, S.; Lichtensztejn, Z.; Neufeld, M.; Ali, J.L.; Altman, A.D.; Nachtigal, M.W.; McManus, K.J. The temporal dynamics of chromosome instability in ovarian cancer cell lines and primary patient samples. PLoS Genet 2017, 13, e1006707, doi:10.1371/journal.pgen.1006707.
  20. Cisyk, A.L.; Nugent, Z.; Wightman, R.H.; Singh, H.; McManus, K.J. Characterizing Microsatellite Instability and Chromosome Instability in Interval Colorectal Cancers. Neoplasia 2018, 20, 943-950, doi:10.1016/j.neo.2018.07.007.
  21. Cisyk, A.L.; Penner-Goeke, S.; Lichtensztejn, Z.; Nugent, Z.; Wightman, R.H.; Singh, H.; McManus, K.J. Characterizing the prevalence of chromosome instability in interval colorectal cancer. Neoplasia 2015, 17, 306-316, doi:10.1016/j.neo.2015.02.001.
  22. Lepage, C.C.; Palmer, M.C.L.; Farrell, A.C.; Neudorf, N.M.; Lichtensztejn, Z.; Nachtigal, M.W.; McManus, K.J. Reduced SKP1 and CUL1 expression underlies increases in Cyclin E1 and chromosome instability in cellular precursors of high-grade serous ovarian cancer. Br J Cancer 2021, 124, 1699-1710, doi:10.1038/s41416-021-01317-w.
  23. Bungsy, M.; Palmer, M.C.L.; Jeusset, L.M.; Neudorf, N.M.; Lichtensztejn, Z.; Nachtigal, M.W.; McManus, K.J. Reduced RBX1 expression induces chromosome instability and promotes cellular transformation in high-grade serous ovarian cancer precursor cells. Cancer Lett 2021, 500, 194-207, doi:10.1016/j.canlet.2020.11.051.

Round 2

Reviewer 2 Report

Major Comments for authors:

1.     Indeed, the authors used three cell lines but only one of which is malignant (HCT116). The references which the authors quoted (reference 1-6) to imply that HCT116 is used widely in an extensive array of CIN-based studies were the authors’ previous works i.e. self-citations. It is not apparent whether reference 7 is a book chapter or journal article and hence cannot be independently verified.  While this reviewer appreciates the authors’ clarification that the cell lines chosen have to be karyotypically stable for CIN studies, and that the choice of HCT116 is for ease of identification of CIN genes since the mutator phenotype of the cell line may synergize with or amplify the CIN phenotype, this should be clearly stated in the Methods section (2.1). Did the authors try to find an appropriate microsatellite-stable and karyotypically stable cell line to replicate the experiments? If this cannot be found, it should also be stated. Similarly, are the non-malignant/early transformed cell lines i.e. 1CT and A1309 microsatellite-stable? 

2  2. The authors stated that ‘Although not the focus of the current study, similar health outcome analyses were performed for CRC patient samples with copy number gains and mutations. No statistical differences (p-values >0.05) in survival outcomes occurred when comparing cases with SKP2 copy number gains or SKP2 mutations to diploid cases’. If these analyses were the works of other researchers, please provide references. If these analyses were the works of the authors, this information should be included in the Discussion section as this is a contrasting outcome of interest.

Minor Comments for authors:

    1.  The authors should mention in the legend for Fig. S5 and S6 that the expected SKP2 expression levels are after 20 passages of the cell lines.

    2.    The pages and lines quoted in the Authors’ reply (e.g., page 9, lines 378-82) do not correspond to that in the manuscript (page 11, lines 366-370).

Author Response

Major Comments for authors:

1. Indeed, the authors used three cell lines but only one of which is malignant (HCT116). The references which the authors quoted (reference 1-6) to imply that HCT116 is used widely in an extensive array of CIN-based studies were the authors’ previous works i.e., self-citations. It is not apparent whether reference 7 is a book chapter or journal article and hence cannot be independently verified.  

     We do not dispute that the previous references were self-citations, and were included as they directly employ HCT116 in similar CIN-based assays. Noting the Reviewers concern, and to be more inclusive, we have not included two additional citations that are completely independent of our own work and are from the laboratory of Dr. Bert Vogelstein from Johns Hopkins University:

  1. Cahill, D.P.; Lengauer, C.; Yu, J.; Riggins, G.J.; Willson, J.K.; Markowitz, S.D.; Kinzler, K.W.; Vogelstein, B. Mutations of mitotic checkpoint genes in human cancers. Nature 1998, 392, 300-303, doi:10.1038/32688.
  2. Rajagopalan, H.; Jallepalli, P.V.; Rago, C.; Velculescu, V.E.; Kinzler, K.W.; Vogelstein, B.; Lengauer, C. Inactivation of hCDC4 can cause chromosomal instability. Nature 2004, 428, 77-81, doi:10.1038/nature02313

     We also thank the Reviewer for highlighting our erroneous inclusion of “reference 7”, which has now been removed. Finally, we amended the manuscript to include mention of the new references (Page 3; Lines 139-141), “Additionally, all three cell lines, and HCT116 in particular, have been employed in a number of previous CIN-based studies[1-8]”.

2. While this reviewer appreciates the authors’ clarification that the cell lines chosen have to be karyotypically stable for CIN studies, and that the choice of HCT116 is for ease of identification of CIN genes since the mutator phenotype of the cell line may synergize with or amplify the CIN phenotype, this should be clearly stated in the Methods section (2.1).

     As suggested by the Reviewer, we have now included the following statements within the Materials & Methods section (Page 3; Lines 116-118),  “To evaluate CIN within a CRC context, we purposefully chose three karyotypically stable colonic epithelial cell lines in which to evaluate the impact reduced SKP2 expression has on CIN phenotypes”; and,

(Page 3; Lines 122-126), “HCT116 is a microsatellite instability cell line that contains a MutL Homolog 1 (MLH1) deficiency underlying defects in DNA mis-match repair [48]. Thus, these cells exhibit a mutator phenotype that may produce mutations capable of synergizing with or amplifying CIN phenotypes that arise following reduced SKP2 expression.”

3. Did the authors try to find an appropriate microsatellite-stable and karyotypically stable cell line to replicate the experiments? If this cannot be found, it should also be stated. Similarly, are the non-malignant/early transformed cell lines i.e. 1CT and A1309 microsatellite-stable? 

     We are unaware of any malignant colonic epithelial cell lines that are both microsatellite stable and karyotypically stable (i.e., exhibit neither MSI nor CIN). In general, CRC cell lines tend to be either MSI or CIN, and most are predominantly CIN. Based on the Reviewer’s suggestion, we have now amended the Materials & Methods section as follows (Page 3; Lines 131-132), “To our knowledge, there are no malignant colonic epithelial cell lines that are both microsatellite stable and karyotypically stable”.

     The MSI status of the 1CT and A1309 cell lines was not assessed by the original authors that generated the cell lines; however, as these cell lines were derived from normal colonic epithelial tissues from a healthy male patient during routine colonoscopy [9,10], they are not suspected to harbor an MSI phenotype.

4. The authors stated that ‘Although not the focus of the current study, similar health outcome analyses were performed for CRC patient samples with copy number gains and mutations. No statistical differences (p-values >0.05) in survival outcomes occurred when comparing cases with SKP2 copy number gains or SKP2 mutations to diploid cases’. If these analyses were the works of other researchers, please provide references. If these analyses were the works of the authors, this information should be included in the Discussion section as this is a contrasting outcome of interest.

     The SKP2 copy number losses, gains and mutation analyses were performed by the authors on patient data obtained from The Cancer Genome Atlas (TCGA) PanCancer Atlas using cBioPortal as described in Materials and Methods Section 2.3 (Page 4; Lines 153-160).

     As recommended by the Reviewer, we have included the following statement within the manuscript (Page 5; Lines 269-274), "Although not the focus of the current study, similar health outcome analyses were performed for CRC patient samples with copy number gains and mutations. No statistical differences (p-values >0.05) in survival outcomes were observed when comparing cases with either SKP2 copy number gains or SKP2 mutations to relative to diploid cases[11]. Thus, based on these analyses, only SKP2 copy number losses are associated with statistically worse outcomes.”

     Finally, and as recommended by the Reviewer, we have also now included the following statement within the Discussion (Page 14; Lines 554-556): “…, while worse survival outcomes were not observed for patient samples with SKP2 copy number gains or mutations.”

Minor Comments for authors:

1. The authors should mention in the legend for Fig. S5 and S6 that the expected SKP2 expression levels are after 20 passages of the cell lines.

     Both Supplementary Figure legends have been amended as suggested and the passage numbers (p20) have now been included in each figure legend.

2. The pages and lines quoted in the Authors’ reply (e.g., page 9, lines 378-82) do not correspond to that in the manuscript (page 11, lines 366-370).

     We apologize for the discrepancies, some of which are our errors and some of which may be due to formatting differences associated with different versions of Word or the use of Letter vs. A4 page formatting. In any case, we have provided all the specific changes in quotes, which can also be identified in the submitted manuscript using the search function or looking for the tracked changes. We apologize for any confusion this may have caused.

REFERENCES:

  1. Thompson, L.L.; McManus, K.J. A novel multiplexed, image-based approach to detect phenotypes that underlie chromosome instability in human cells. PLoS One 2015, 10, e0123200, doi:10.1371/journal.pone.0123200.
  2. Thompson, L.L.; Baergen, A.K.; Lichtensztejn, Z.; McManus, K.J. Reduced SKP1 expression induces chromosome instability through aberrant cyclin E1 protein turnover. Cancers 2020, 12, 531-531, doi:10.3390/cancers12030531.
  3. Asbaghi, Y.; Thompson, L.L.; Lichtensztejn, Z.; McManus, K.J. KIF11 silencing and inhibition induces chromosome instability that may contribute to cancer. Genes Chromosomes Cancer 2017, 56, 668-680, doi:10.1002/gcc.22471.
  4. Guppy, B.J.; McManus, K.J. Mitotic accumulation of dimethylated lysine 79 of histone H3 is important for maintaining genome integrity during mitosis in human cells. Genetics 2015, 199, 423-433, doi:10.1534/genetics.114.172874.
  5. Barber, T.D.; McManus, K.; Yuen, K.W.; Reis, M.; Parmigiani, G.; Shen, D.; Barrett, I.; Nouhi, Y.; Spencer, F.; Markowitz, S.; et al. Chromatid cohesion defects may underlie chromosome instability in human colorectal cancers. Proc Natl Acad Sci U S A 2008, 105, 3443-3448, doi:10.1073/pnas.0712384105.
  6. Palmer, M.C.L.; Neudorf, N.M.; Farrell, A.C.; Razi, T.; Lichtensztejn, Z.; McManus, K.J. The F-box protein, FBXO7 is required to maintain chromosome stability in humans. Hum Mol Genet 2021, doi:10.1093/hmg/ddab330.
  7. Cahill, D.P.; Lengauer, C.; Yu, J.; Riggins, G.J.; Willson, J.K.; Markowitz, S.D.; Kinzler, K.W.; Vogelstein, B. Mutations of mitotic checkpoint genes in human cancers. Nature 1998, 392, 300-303, doi:10.1038/32688.
  8. Rajagopalan, H.; Jallepalli, P.V.; Rago, C.; Velculescu, V.E.; Kinzler, K.W.; Vogelstein, B.; Lengauer, C. Inactivation of hCDC4 can cause chromosomal instability. Nature 2004, 428, 77-81, doi:10.1038/nature02313.
  9. Roig, A.I.; Eskiocak, U.; Hight, S.K.; Kim, S.B.; Delgado, O.; Souza, R.F.; Spechler, S.J.; Wright, W.E.; Shay, J.W. Immortalized epithelial cells derived from human colon biopsies express stem cell markers and differentiate in vitro. Gastroenterology 2010, 138, 1012-1021 e1011-1015, doi:10.1053/j.gastro.2009.11.052.
  10. Zhang, L.; Kim, S.; Jia, G.; Buhmeida, A.; Dallol, A.; Wright, W.E.; Fornace, A.J.; Al-Qahtani, M.; Shay, J.W. Exome Sequencing of Normal and Isogenic Transformed Human Colonic Epithelial Cells (HCECs) Reveals Novel Genes Potentially Involved in the Early Stages of Colorectal Tumorigenesis. BMC Genomics 2015, 16 Suppl 1, S8, doi:10.1186/1471-2164-16-S1-S8.
  11. Hoadley, K.A.; Yau, C.; Hinoue, T.; Wolf, D.M.; Lazar, A.J.; Drill, E.; Shen, R.; Taylor, A.M.; Cherniack, A.D.; Thorsson, V.; et al. Cell-of-Origin Patterns Dominate the Molecular Classification of 10,000 Tumors from 33 Types of Cancer. Cell 2018, 173, 291-304 e296, doi:10.1016/j.cell.2018.03.022.